# Causal effects of population dynamics and environmental changes on spatial variability of marine fishes

Jheng-Yu Wang [1], Ting-Chun Kuo [1,2] & Chih-hao Hsieh [1,3,4,5] ✉

Populations with homogeneous distributions have better bet-hedging capacity than more heterogeneously distributed populations. Both population dynamics and environmental factors may influence the spatial variability of a population, but clear empirical evidence of such causal linkages is sparse. Using 25-year fish survey data from the North Sea, we quantify causal effects of age structure, abundance, and environment on nine fish species. We use empirical dynamic modeling—an approach based on state-space reconstruction rather than correlation—to demonstrate causal effects of those factors on population spatial variability. The causal effects are detected in most study species, though direction and strength vary. Specifically, truncated age structure elevates population spatial variability. Warming and spatially heterogeneous temperatures may enhance population spatial variability, whereas abundance and large-scale environmental effects are inconclusive. Fishing may affect population spatial variability directly or indirectly by altering age structure or abundance. We infer potential harmful effects of fishing and environmental changes on fish population stability, highlighting the importance of considering spatial dynamics in fisheries management.

[1] Institute of Oceanography, National Taiwan University, No.1, Section 4, Roosevelt Road, Taipei 10617, Taiwan. [2] Institute of Marine Affairs and Resources Management, National Taiwan Ocean University, No.2, Peining Road Jhongjheng District, Keelung City 20224, Taiwan. [3] Institute of Ecology and Evolutionary Biology, Department of Life Science, National Taiwan University, No.1, Section 4, Roosevelt Road, Taipei 10617, Taiwan. [4] Research Center for Environmental Changes, Academia Sinica, No.128, Section 2, Academia Road Nankang, Taipei 11529, Taiwan. [5] National Center for Theoretical Sciences, No.1, Section 4, Roosevelt Road, Taipei 10617, Taiwan. ✉email: chsieh@ntu.edu.tw

Spatial structure is critical for a population to buffer environmental fluctuations. By distributing broadly and homogeneously in space (i.e., with low population spatial variability), species can spread risks of extinction over various habitats when encountering environmental changes[1]; this strategy is known as bet-hedging. However, species naturally distribute heterogeneously and adjust their spatial structure in response to population dynamics[2] and environmental variability[2,3]. For example, changes in age structure can affect population spatial variability[4,5], since different age classes live in different habitats according to age-specific living requirements, mobility, and competitive advantages[6]. Hence, populations with a diverse age structure are more capable of occupying various habitats and should have lower population spatial variability[5,7]. Besides, increased abundance may lead to lower population spatial variability[8], as a growing population typically expands from optimal habitats to suboptimal habitats to ease intraspecific competition[9]. Other than population dynamics, environmental disturbances (e.g., changes in temperature) also shape population spatial variability by inducing various survival rates in different habitats[7,10].

Although relationships between population spatial variability versus age structure, abundance, and environment have been described[1,6], to our knowledge, no study has systematically quantified causal effects of these factors on population spatial variability, especially for marine ecosystems. Existing analyses on determining causation are generally based on linear approaches (e.g., correlation, regression, or structure equation modeling) that can yield ambiguous results in complex-interdependent dynamical systems, which are common in nature[11,12]. Here, we employ convergent cross mapping[12] (CCM) and S-map[13], two recently developed approaches specifically designed for nonlinear dynamical systems, to quantitatively determine causal relationships. These methods, in contrast to linear approaches, can distinguish causality from correlation by depicting underlying mechanisms of a dynamical system[12] (see "Methods" for details).

We quantitatively measure causal effects of age structure, abundance, and environment on population spatial variability, using a 25-year (1991–2015) biquarterly dataset of nine exploited fish species (Supplementary Table 1) from the International Bottom Trawl Survey (IBTS) in the North Sea. The population spatial variability is measured by the coefficient of variation (CV) of abundance across spaces (see "Methods" section). We test three hypotheses: (1) increasing age diversity reduces population spatial

variability because a population with a diverse age structure can occupy various habitats[5,7]; (2) increasing abundance reduces population spatial variability because a population tends to expand its occupancy when abundance increases[9]; and (3) warming and spatially heterogenious temperatures increases population spatial variability because unfavorable environmental conditions can reduce local population size[7,10]. To test our hypotheses, we firstly use CCM to identify causal variables of population spatial variability, with consideration of potential lagged effects. Next, we choose the most critical causal variables and estimate their quantitative causal effects (direction and influential strength) on population spatial variability at each time step using S-map (see "Methods"). We also explore potential causal effects of fishing mortality on population spatial variability and confounding factors.

We identify that age diversity, abundance, and examined environmental variables have causal effects on population spatial variability for most study species using CCM. Our S-map results suggest that decreasing age diversity enhances population spatial variability. However, changes in abundance either increase or decrease population spatial variability, which might depend on aggregation tendency of the species. Warming and spatially heterogeneous temperatures tend to enhance population spatial variability. Fishing may affect the population spatial variability, either directly or indirectly by altering age structure or abundance. These findings link population spatial structure with population dynamics and environment for marine fish species. Our results highlight the importance of considering population spatial dynamics in stock assessments and fisheries management.

## Results and discussion
**Causal variables of population spatial variability.** Based on our CCM analysis, spatial variability of each species responded to nearly all examined variables (Table 1). We presented empirical evidence for causal effects of age diversity, abundance, and environmental variables on population spatial variability from a perspective of dynamical system instead of correlation. These results were supported by previous observations that population spatial structure adjusted in response to changes in age structure, abundance, and environmental conditions[2,6,14]. Nevertheless, the detected causal effects of examined variables on population spatial variability were in general weak or moderate (Table 1). Such a weakly linked complex causal network creates difficulties for

---

**Table 1 Causal effect of examined variables on population spatial variability.**

| Species | Common name | Dimensionality (E*) | Library variable: spatial CV of CPUE | | | | |
|---|---|---|---|---|---|---|---|
| | | | Age diversity | Abundance | AMO | Temperature | CV of temperature |
| *Clupea harengus* | Atlantic herring | 5 | 0.1796 (1) | 0.0801 (1) | 0.2801 (1) | n.s. | 0.2650 (5) |
| *Gadus morhua* | Atlantic cod | 6 | 0.4311 (1) | 0.4269 (0) | 0.5426 (0) | 0.9168 (0) | 0.7743 (0) |
| *Melanogrammus aeglefinus* | Haddock | 5 | 0.1989 (1) | 0.3080 (3) | 0.4653 (1) | n.s. | 0.2644 (6) |
| *Merlangius merlangus* | Whiting | 4 | 0.2965 (4) | 0.2094 (2) | 0.3767 (4) | n.s. | n.s. |
| *Pleuronectes platessa* | Plaice | 6 | 0.3363 (1) | 0.1744 (1) | 0.4648 (6) | 0.8791 (6) | 0.7908 (7) |
| *Pollachius virens* | Saithe | 2 | 0.0911 (2) | 0.2703 (5) | 0.1721 (1) | 0.7269 (6) | 0.6836 (5) |
| *Scomber scombrus* | Atlantic mackerel | 6 | 0.6309 (3) | 0.1610 (8) | 0.3101 (5) | 0.9901 (0) | 0.7468 (7) |
| *Sprattus sprattus* | Sprat | 7 | 0.1846 (3) | 0.1944 (3) | 0.1765 (2) | 0.6039 (4) | n.s. |
| *Trisopterus esmarkii* | Norway pout | 3 | 0.3935 (4) | 0.3197 (4) | 0.0620 (7) | 0.1834 (2) | 0.3052 (0) |

Values in each variable column indicate causal effect of the variable on library variable. Larger values indicate stronger causal effects. Numbers in brackets indicate lag at which causal effect was strongest for the corresponding variable. Causal effect was determined by convergence cross mapping (CCM; see "Methods"). In dynamical theory, if a time-series variable $X(t)$ has causal effect on another time-series variable $Y(t)$, one can predict the shadow manifold of $X(t)$ using that of $Y(t)$. Here, we used shadow manifold of library variable (i.e., population spatial variability) to predict shadow manifold of each examined variable. Predictive ability was measured by correlation coefficient ($\rho$) between predicted and observed data, and can be an indicator of causal effect. Only significant resutls were shown ($p < 0.05$ in both one-sided Kendall's $\tau$ test and Student's $t$-test on $\rho$).
E* optimal embedding dimension of library variables, CV coefficient of variation, CPUE catch per unit effort, AMO Atlantic Multidecadal Oscillation, n.s. nonsignificant causal effect.

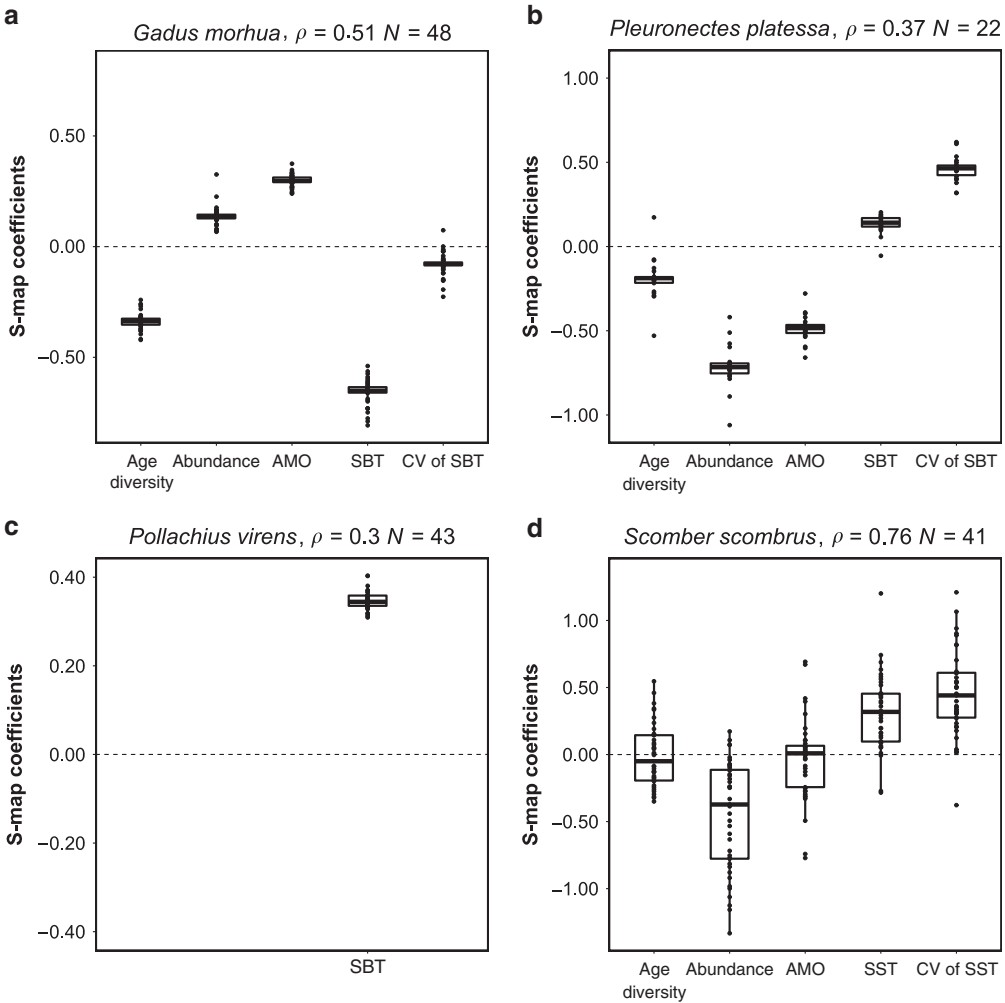

**Fig. 1 Boxplot of overtime influential strengths of selected causal variables on population spatial variability.** Influential strengths were coefficients of S-map model estimated at each time step (see "Methods"). A positive S-map coefficient indicates a positive causal effect of the variable on population spatial variability, and vice versa. Magnitude of S-map coefficient represents strength of causal effect. $\rho$ indicates performance of S-map model. Significant S-map results were detected for **a** Atlantic cod, **b** plaice, **c** saithe, and **d** Atlantic mackerel ($p < 0.1$, one-sided Student's $t$-test on $\rho$). Causal variables were selected according to embedding dimension and their rank of causal effects determined by CCM (see Table 1). AMO indicates Atlantic Multidecadal Oscillation. SBT and SST indicate sea bottom and sea surface temperatures, respectively. CV indicates coefficient of variation. The bold line represents the median. The lower/upper hinges correspond to the first/third quartiles. The lower/upper whisker extends from the hinge to the smallest/largest value no further than 1.5 times of interquartile range. Data are shown as dots.

traditional linear approaches, e.g., Granger causality, to decipher causal information among time series[12].

**Overtime influential strengths of causal variables**. After identifying causal variables, we quantified overtime influential strengths of age diversity, abundance, and environmental variables on population spatial variability using S-map. S-map is an algorithm that can reconstruct the underlying attractor describing interactions among variables (see "Methods" for details of reconstructing attractor governing the dynamical system). We identified four of the nine study species having significant attractor reconstruction (Fig. 1, Table 2). The remaining five species either had nonsignificant attractor reconstruction (Supplementary Table 2) or a lack of causal variables to reconstruct the attractor (i.e., sprat (*S. sprattus*)). It should be noted that for the species having nonsignificant attractor reconstruction in S-map analysis did not mean that CCM-determined causal variables were invalid. Instead, we inferred that in these species, some other important causal variables not included in our analyses were

critical for reconstructing the attractor (see "Methodological considerations" below). Based on our S-map results, we observed several important patterns. First, for species with significant attractor reconstruction that included age diversity (i.e., Atlantic cod (*Gadus morhua*), plaice (*Pleuronectes platessa*), and Atlantic mackerel (*Scomber scombrus*)), the overtime causal effects of age diversity on population spatial variability were on average negative (Fig. 1a, b, d), which supported our first hypothesis. Second, abundance overall negatively affected population spatial variability for plaice and Atlantic mackerel, supporting our second hypothesis (Fig. 1b, d); however, an average of positive abundance effect was observed for Atlantic cod (Fig. 1a). Third, temperature and/or its spatial variability overall positively affected population spatial variability for three species with significant S-map reconstruction (Fig. 1b–d), which agreed with our third hypothesis; in contrast, both temperature and spatial variability of temperature had an average negative effect on spatial variability of Atlantic cod (Fig. 1a). Finally, the Atlantic Multidecadal Oscillation (AMO) had a positive effect for Atlantic cod (Fig. 1a), but on average had a negative effect for plaice and Atlantic

**Table 2 Average overtime influential strengths of selected causal variables on population spatial variability.**

| Species | Library variable: CV of CPUE | | | | | $\theta$ | $\rho$ | *p*-Value |
|---|---|---|---|---|---|---|---|---|
| | Age diversity | Abundance | AMO | Temperature | CV of temperature | | | |
| *Gadus morhua* (Atlantic cod) | −0.3372 | 0.1370 | 0.2990 | −0.6565 | −0.0796 | 0 | 0.5100 | <0.001 |
| *Pleuronectes platessa* (Plaice) | −0.1928 | −0.7130 | −0.4860 | 0.1340 | 0.4592 | 0 | 0.3705 | 0.0448 |
| *Pollachius virens* (Saithe) | | | | 0.3470 | | 0 | 0.3031 | 0.0241 |
| *Scomber scombrus* (Atlantic mackerel) | −0.0114 | −0.4519 | −0.0332 | 0.3111 | 0.4501 | 3 | 0.7568 | <0.001 |

Influential strengths were estimated by S-map model at each time point during study period (see "Methods"), and were averaged over time. Causal variables were selected according to embedding dimension and their rank of causal effects determined by convergence cross mapping (CCM; see Table 1). Only species with significant S-map results were shown (*p* < 0.1, one-sided Student's *t*-test on $\rho$).
CV coefficient of variation, CPUE catch per unit effort, AMO Atlantic Multidecadal Oscillation, $\theta$ nonlinearity of the dynamical system. If $\theta = 0$, S-map model reduces to a linear vector autoregressive model[40]; $\rho$ performance of S-map model.

mackerel (Fig. 1b, d). Note that the estimated influential strengths of selected causal variables varied through time for all species (Supplementary Fig. 1), highlighting the nonlinear context-interdependent behavior in dynamical systems[15].

There are multiple ways to reconstruct the attractor for a dynamical system using different combinations of causal variables[16,17]. To test the robustness of aforementioned patterns, we used various alternative combinations of CCM-determined causal variables to reconstruct the attractor for dynamics of population spatial variability and repeated S-map analyses (see "Methods"). We additionally detected one significant S-map result (see Supplementary Table 3 for saithe (*Pollachius virens*)), when including spatial variability of temperature as a causal variable. This result indicated that the spatial variability of temperature positively affected population spatial variability for saithe, which again supported our third hypothesis. In the following parts, we discuss the influential strength of each causal variable on population spatial variability.

**Age diversity effects**. Our dynamical approaches provided empirical quantification that age diversity negatively affected population spatial variability (Table 2, Supplementary Table 3). That is, the more diverse age structure (i.e., higher age diversity) a population had, the more homogeneous spatial structure (i.e., lower spatial variability) the population manifested (Fig. 2). This causal relationship was supported by the observation that different age classes lived in different habitats (Fig. 3, see also Supplementary Fig. 2). The observed age-dependent spatial distribution, also reported in other marine species[18–20], suggested that a diverse age structure enabled a population to occupy diverse habitats, and to form a broad and homogeneous spatial structure. Such homogeneous spatial structure provided bet-hedging capacity for a species to alleviate harmful effects of environmental disturbances at local habitats. Therefore, maintaining a complete and diverse age structure would be regarded as critical for a population to survive in a changing environment.

**Abundance effects**. Our second hypothesis that increasing abundance reduces population spatial variability was observed in plaice and Atlantic mackerel (Table 2, see also Supplementary Fig. 3); this partially supported the observation in the previous study[7]. The negative causation between abundance and population spatial variability can be explained by the density-dependent habitat selection theory. When abundance is low, the population will aggregate at specific preferred habitats[9], resulting in a higher population spatial variability. However, as abundance increases to a certain level at which the stress of intraspecific competition becomes significant, the population will start moving into less

optimal habitats to maximize overall fitness[21], thereby reducing spatial variability of the population.

However, increasing abundance enhanced spatial variability of Atlantic cod (Table 2, see also Supplementary Fig. 4). We hypothesize that the relationship between abundance and population spatial variability might be determined by the aggregation tendency of a species, which can be measured by Taylor's exponent. In Taylor's power law[22], the mean ($M$) and variance ($V$) of population abundance across spaces are exponentially related: $V = aM^b$. The exponent $b$ represents aggregation tendency of the population[5,23]; therefore, a larger Taylor's exponent indicated that a population would be more aggregated when its abundance increases. With algebraic manipulations, we can manipulate the equation as $V^{\frac{1}{2}}M^{-1} = CV = a'M^{\frac{b}{2}-1}$, where CV is coefficient of variation (spatial variability) and $a'$ is a constant. If $b$ falls between one and two, population spatial variability (CV) is negatively proportional to abundance ($M$), as described in our second hypothesis. Contrastly, if $b$ is larger than two, population spatial variability is instead positively proportional to abundance. Based on the manipulated Taylor's equation, we thus propose that the direction of the abundance effects on population spatial variability depends on the aggregation tendency of a population. If a species has a stronger aggregation tendency (i.e., $b > 2$), its spatial variability should enhance as abundance increases, and vice versa. Our results, though with limited number of species, supported this hypothesis. We identified a positive abundance effect on population spatial variability for Atlantic cod, which had $b > 2$, whereas the effect became negative for plaice and Atlantic mackerel, which had $b < 2$ (see abundance effect in Table 2 and Taylor's exponent in Supplementary Table 1).

Our hypothesis of aggregation tendency provided a theoretical explanation for previous contradictory observations on the relationship between population spatial structure and abundance. For example, some species (e.g., Atlantic mackerel) had an extended spatial distribution in response to the increasing stock size[24], as suggested by density-dependent process. However, some other species (e.g., cod) instead had a concentrated spatial distribution when stock size increased[25], probably because aggregation may benefit the population spawning[26]. Our results implied that response of spatial variability of fish species to changing abundance may be regulated by both density-dependent process and behavioral aggregation tendency. Certainly, more data are needed to investigate this hypothesis.

**Environmental effects**. Increasing population spatial variability with warming temperatures was observed in plaice, saithe, and Atlantic mackerel (Table 2, see also Supplementary Fig. 5), which supported our third hypothesis. The resulting heterogeneous

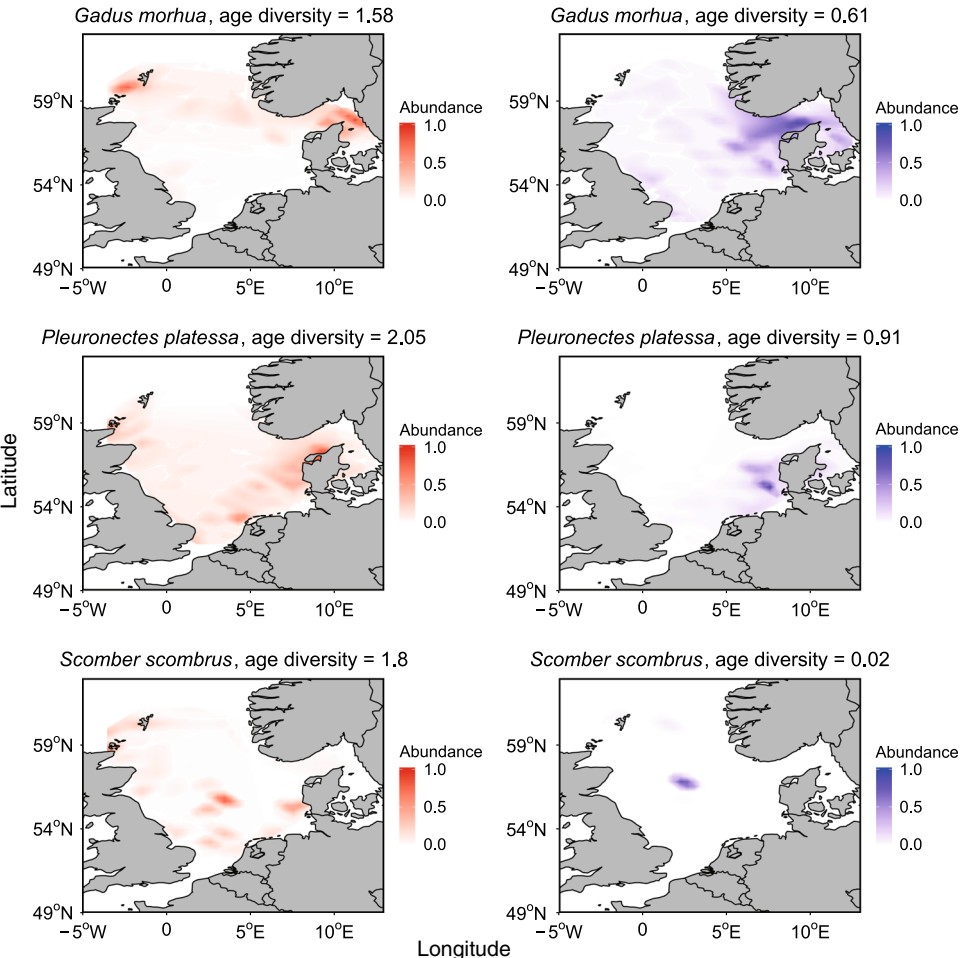

**Fig. 2 Population spatial distribution at the time when age diversity was highest (left) versus lowest (right).** Populations tended to distribute more evenly in space when their age diversity was higher. Only species with significant S-map results involving age diversity were shown ($p < 0.1$, see Table 2).

spatial distribution may be because of local reduction or local extinction of populations that were not adapted to warming temperatures. This phenomenon was especially apparent for species that moved less with changing temperatures, e.g., plaice and saithe[10]. On the contrary, warming temperatures decreased spatial variability of Atlantic cod (Table 2, Supplementary Fig. 6), probably due to their relatively significant migratory ability in response to changing temperatures[10,27].

In addition, spatially heterogeneous temperatures increased population spatial variability for plaice, Atlantic mackerel, and saithe (Table 2, Supplementary Table 3). As temperature restricts living areas of fish species, spatially heterogeneous temperatures may force a population to form a heterogeneous spatial distribution (Supplementary Fig. 7). Spatially heterogeneous temperatures can also reduce survival of larvae and juveniles, when they move from a spawning ground to a nursery ground[28], leading to fragmented spatial distribution of the population. Nonetheless, cod still exhibited a more homogeneous spatial structure in response to a more spatially heterogeneous temperature (Table 2, see also Supplementary Fig. 8). We inferred that their stronger movement in response to changing temperature[10] might enable them to inhabit suitable habitats more homogeneously.

In the present study, AMO had no consistent casual effect on population spatial variability among species (Table 2). As a large-scale environmental indicator, AMO represented outcomes of complex interactions between atmosphere and ocean circulation systems; therefore, the mechanism of AMO effect on population spatial variability may not be straightforward[29]. It has been

documented that spatial distribution and abundance of fish populations may be linked with AMO (refs. [2,30]). Plankton biomass and community, critical in determining survival, as well as spatial distribution of fish larvae and recruitment[31], are also associated with AMO (ref. [30]). Interactive effects of these factors linked with AMO complicated the ultimate causal effect of AMO on spatial variability of fish populations, which may have contributed to the ambiguous causal effects in this study.

**Fishing and confounding causal effects.** Considering that overfishing is a critical driver of age truncation and abundance reduction[1], we also examined whether and how fishing mortality could indirectly affect population spatial variability by altering age structure and/or abundance. To do so, we explored causal effects of fishing mortality on age diversity and abundance, respectively, and estimated the corresponding influential strengths. These analyses could only be done at a yearly base in accordance with annual fishing mortality data (see "Methods").

Using CCM, we determined that fishing mortality causally affected age diversity and/or abundance in five of the nine study species (see Supplementary Tables 4 and 5 for causal effect of fishing mortality on Atlantic cod, whiting (*Merlangius merlangus*), plaice, saithe, and Norway pout (*Trisopterus esmarkii*)). The detected causal effects for age diversity and abundance were less significant compared to that for population spatial variability (see CCM results in Table 1 versus Supplementary Tables 4 and 5). This was attributed to insufficient time-series length when the

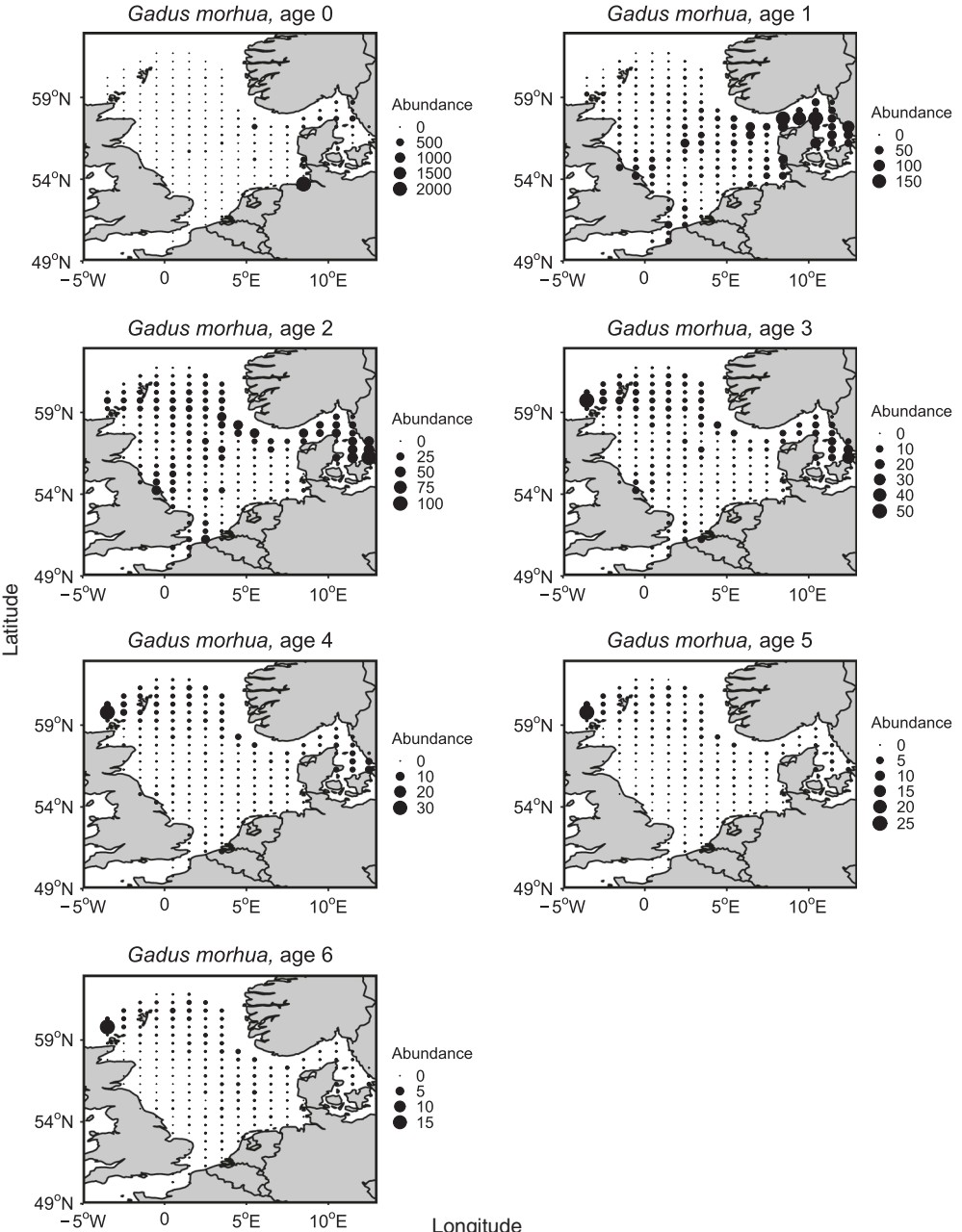

**Fig. 3 Atlantic cod (G. morhua) as an example illustrating that different age classes had different spatial distributions.** Each circle indicates average abundance at a survey location during study period. Size of circles increases with abundance. Younger cod (age classes 0–3) were abundant in the Skagerrak (the western North Sea), whereas older cod (age classes 4–6) were abundant in the northwest of the North Sea. Other species also had age-dependent property in their spatial distributions (Supplementary Fig. 2).

data were shortened to be yearly base, causing a difficulty in reconstructing the attractor. We further estimated the influential strengths of CCM-detected causal variables on age diversity and abundance using S-map, and averaged the influential strengths over all significant lags of each causal variable to reduce variability arising from short time series. In these S-map results, fishing mortality had a negative effect on age diversity for Atlantic cod and whiting (Supplementary Table 6), which partially agreed with previous observations on age truncation associated with fishing[1], although the effect was on average slightly positive for plaice. In addition, increasing fishing mortality was detected to reduce the population abundance for Atlantic cod, whiting, and saithe (Supplementary Table 7). The observed causal effects of fishing mortality on age diversity/abundance suggested that

fishing may undermine population spatial structure (e.g., enhanced population spatial variability) via age truncation and abundance reduction, given the identified negative causal effects of age diversity and abundance on population spatial variability. This could account for exploited species being more vulnerable to environmental disturbance[4,7,32], as fishing-induced heterogeneous spatial structure might weaken bet-hedging capacity of these species[1]. Even worse, because responses of population spatial variability to changing age diversity and abundance were generally time-delayed (see ubiquitous-lagged causal effects in Table 1), an undermined spatial structure may have a prolonged recovery, even if age structure and abundance are restored.

We also included fishing mortality as a potential causal variable of population spatial variability to test if fishing could directly

affect population spatial variability. According to CCM, fishing mortality was a significant causal variable in five of the nine study species (Supplementary Table 8). However, only Atlantic cod and whiting had significant S-map results, with contradictory fishing mortality effects (Supplementary Table 9). The nonsignificant results of S-map analyses were attributed to the short time series (only 25 time points) that cannot be used to successfully reconstruct the attractor. Nevertheless, working within constraints of limited length of time series, there were indications that fishing mortality might affect population spatial variability, either through a direct path or an indirect path associated with age structure and/or abundance.

It should be noted that causal effects of fishing, age structure, abundance, and environmental variables on population spatial variability were intertwined, reflecting the complexity of the system. Taking Atlantic mackerel as an example, population abundance positively affected age diversity (Supplementary Table 6), whereas age diversity negatively affected population spatial variability (Table 2). Therefore, it is not surprising that abundance was detected to negatively affect population spatial variability (Table 2). Similarly, we detected a negative causal effect of temperature and spatial variability of temperature on age diversity of Atlantic mackerel (Supplementary Table 6), and that age diversity negatively affected population spatial variability (Table 2). As such, both temperature and spatial variability of temperature positively affected population spatial variability (Table 2). Our findings indicated that changes in age structure, abundance, or environmental variables affected not only population spatial structure but also population dynamics, resulting in a confounding causal network among these variables. We also noted that other causal variables not explored in this study might affect population spatial variability. For instance, according to variance–mass allometry[33], populations with smaller mean body mass may have greater variance in their distributions. Since body size of marine fishes may reduce as a result of overfishing[1] and warming temperatures[34], exploited stocks are likely to become more spatially variable and vulnerable to climate variability in a warming environment[32].

**Methodological considerations**. Although we determine causal relationships and demonstrated quantitatively influential strengths of age diversity, abundance, and environmental variables on spatial variability of marine fishes in the North Sea, some caution is warranted when interpreting the results. First, the gear used in IBTS was specifically designed for demersal species and may not catch great quantities of pelagic species. In particular for Atlantic mackerel, mainly relatively young individuals were caught in IBTS, and thus the IBTS data may not represent the whole stock (C. Needle, personal communication). Although this limits the use of IBTS data to estimate the whole population abundance for some species, it should have a less effect on the findings of our study. At least, the interactions among population spatial variability, age diversity, abundance, and environmental variables uncovered by our methods were valid for that proportion of the population we examined. Given that IBTS has a long survey period and large spatial coverage, several studies have shown the potential of using IBTS to study changes in population spatial structure for pelagic species[35–37].

Second, the examined causal relationships in this study were in general not strong for all species (see Table 1 for moderate causal effects), likely due to the limitation of time-series length. Also, dynamics of population spatial variability cannot be fully explained by age diversity, abundance, and selected environmental variables (see Table 2 for moderate performance ($\rho$) of S-map results). This may suggest that some other important

processes influencing population spatial structure were not captured in our study. While we detected many significant causal variables for population spatial variability using CCM (Table 1), for some species, we could not successfully reconstruct the attractor of population spatial variability, using these causal variables in S-map analysis (only four of the eight species had significant S-map results, and one other species had no sufficient causal variables to reconstruct the attractor; see Table 1 and Supplementary Table 2). This was not because the CCM-determined causal variables were invalid, but because there existed some other key variables dominating dynamics of population spatial variability[38]. Therefore, using only the causal variables examined in the present study to reconstruct the attractor and to depict dynamics of population spatial variability was not sufficient. For example, consider a hypothetical causal network—spatial variability of a fish species is causally affected by age diversity, temperature, and prey abundance. Suppose we have only data for age diversity and temperature, but lack data for prey abundance. In such a case, CCM may still successfully identify age diversity and temperature each as one of the causal variables of population spatial variability. This is because the lagged coordinate embedding constructed by a single time series (e.g., age diversity or temperature) allows CCM to recover missing information of a dynamical system (see "Methods"). However, it is not possible for S-map (multivariate embedding) to successfully reconstruct the attractor using only age diversity and temperature, because the third critical variable, i.e., prey abundance, is missing. When characteristics (e.g., geometric shape) of the attractor are mainly determined by the dominant causal variables, using other causal variables to reconstruct the attractor can only partially reveal the underlying dynamical system[38], leading to poor performance of S-map model. Therefore, a further comprehensive examination of causal effects of other biological factors, physical conditions, and ecological events should improve knowledge regarding dynamics of population spatial structure. In addition, a long-term and continuous survey covering a wide range of space and age classes for marine fish species is warranted for a thorough research on population spatial structure associated with population dynamics.

**Final remarks**. Growing evidence has suggested that population spatial structure is key to understanding population dynamics[1,6], and therefore should be included in fisheries management as a complementary indicator to improve monitoring of stock status[39]. Our findings highlighted potentially detrimental effects of fishing on population spatial structure, which may be associated with truncated age structure, as well as diminished abundance. In addition, warming and fluctuating temperatures also had roles in undermining population age structure, abundance, and spatial structure. We therefore emphasize the importance of considering population spatial structure in fishery management, particularly on intertwining effects of changes in population dynamics induced by fishing and environmental variability.

## Methods

**Fish data**. To investigate dynamics of population spatial variability, we explored all spatial–temporal surveys in the Database of Trawl Surveys (DATRAS) of the International Council for the Exploration of the Sea (ICES). We considered only surveys containing age-specific catch data for each species in each grid (i.e., subarea or statistical rectangle with a resolution of 0.5° latitude by 1° longitude defined by ICES). For each species in a survey, to appropriately capture living areas of the species, we removed grids at which catch data of the species was consistently zero throughout the survey period. To avoid biased estimation on population spatial variability, we removed time points in which the number of grids was <10. To ensure viability of empirical dynamic modeling (EDM), we considered species having long enough continuous time-series data (i.e., time-series length $n \geq 30$)[12,40]. Eventually, we identified nine exploited species from the IBTS in the North Sea satisfying our criteria, including *Clupea harengus*, *G. morhua*, *Melanogrammus aeglefinus*, *M. merlangus*,

*P. platessa, P. virens, S. scombrus, S. sprattus*, and *T. esmarkii* (Supplementary Table 1). We used quarterly catch per unit effort (CPUE) data, i.e., the number of individuals caught per hauling hour, in each grid in the North Sea (49.5°–61.5° N; 5° W–13° E) from 1991 to 2015 in this study. Only data in the first and third quarters (Q1 and Q3) were used because sampling frequency was disproportionally low in Q2 and Q4. We further collected biological information for each species, including lifestyle and biogeography information from FishBase (http://www.fishbase.org/search.php) and literature (Supplementary Table 1).

**Population spatial variability**. Population spatial variability was measured using a unitless indicator of variability, namely CV of CPUEs over grids[41]. For CPUE, we used data product "CPUE per length per subarea" of IBTS for all species. We calculated population spatial variability as follows. First, we summed CPUEs across length classes for each subarea and quarter,

$$CPUE_{q,s} = \sum_l CPUE_{q,s,l}, \quad (1)$$

where q, s, and l represent quarter, subarea, and length class, respectively. Thus, a quarterly (Q1 and Q3) time series of total CPUE for each subarea was obtained. Then, we calculated the spatial CV of total CPUE over subareas for each quarter, and obtained a quarterly time series of spatial CV,

$$CV_q = \frac{\sigma_q}{\mu_q}, \quad (2)$$

where $\sigma_q$ and $\mu_q$ are the standard deviation, and mean of $CPUE_{q,s}$ over subareas s on quarter q, respectively.

**Age structure**. We quantified the completion of age structure of each species in the living area using the Shannon index of age class distribution (i.e., age diversity). We used data product "CPUE per age per subarea" of IBTS for all species and calculated age diversity as follows. First, we summed CPUEs across subareas for each age class and quarter,

$$CPUE_{q,a} = \sum_s CPUE_{q,s,a}, \quad (3)$$

where q, s, and a represent quarter, subarea, and age class, respectively. Then, we calculated age diversity for each quarter,

$$Shannon_q = -\sum_a p_{q,a} \ln p_{q,a}, \quad (4)$$

where

$$p_{q,a} = \frac{CPUE_{q,a}}{\sum_a CPUE_{q,a}}. \quad (5)$$

**Abundance**. We calculated total CPUE by summing CPUEs across length classes and subareas for each quarter,

$$CPUE_q = \sum_s \sum_l CPUE_{q,s,l}, \quad (6)$$

where q, s, and l represent quarter, subarea, and length class, respectively.

**Environment**. For environmental variables, we calculated spatial mean and CV of temperatures over the study area for each quarter. We also collected AMO to explore large-scale environmental effects. Regarding temperature, we considered sea surface temperatures (SST) for pelagic species and sea bottom temperatures (SBT) for demersal species, respectively, to appropriately capture causal effects of temperature for species with disparate lifestyles. SBT data were available from ICES, whereas SST and AMO data were collected from National Oceanic and Atmospheric Administration.

**Empirical dynamic modeling**. In the dynamical system theory, a time series is a projection from the motion of the attractor in multidimensional state space onto one coordinate axis of variables associated with the dynamical system (refer to a brief animation on tinyurl.com/EDM-intro). The attractor is a collection of all trajectories and states governing dynamics of the system. In a dynamical system, the state changes according to a set of rules (i.e., dynamical processes). Knowing the process of the attractor is equivalent to knowing the behavior of the projected time series. Although the attractor of a dynamical system is usually unknown a priori, there is an empirical way to reconstruct a shadow attractor manifold that is topologically invariant to the original attractor manifold[42,43]. Specifically, one can reconstruct the shadow manifold under an embedding space consisting of lagged coordinates from a single time series, i.e., <X(t), X(t − τ), …, X(t − (E − 1)τ)>, where τ is the time lag and E is the embedding dimension. The use of lagged coordinates to reconstruct the attractor manifold is also known as state-space reconstruction. Since the reconstructed shadow manifold preserves mathematical properties of the original manifold, several approaches based on state-space reconstruction have been developed to investigate the dynamical system. CCM (ref. [12]) and S-map[13], two tools of EDM built on state-space reconstruction, can

determine causal relationships and estimate quantitatively corresponding causal effects between variables. We explain these two methods in the following sections.

**Identification of causal variables of population spatial variability**. In the theory of dynamical system, two causally linked time-series variables are from the same dynamical system. Both shadow manifolds reconstructed by the two time series are topologically invariant to the original attractor manifold, and therefore can identify the state of each other. If a time-series variable, Y(t), causally affects another variable, X(t), information about the states of Y(t) will be recorded in X(t). As such, one can use the information recorded in X(t) to recover Y(t). Based on this concept, CCM tests causality by measuring the extent to which a causal variable has left an imprint in the time series of an affected variable. The essential ideas are summarized in the following brief animations: tinyurl.com/EDM-intro. Sugihara et al.[12] indicated that "Y(t) causes X(t)" can be determined if points on the shadow manifold of X(t) (library variable) can predict (cross-map to) the contemporaneous points on the shadow manifold of Y(t), and this prediction converges—meaning that the cross-mapping skill ρ(L) improves with increasing library length (L) of X. Here, ρ is the correlation coefficient between observed and predicted data and L is the length of interval randomly subsampled from the time series (i.e., subsampling size used to construct library). Convergence is determined by examining ρ under various L. Here, L starts from the minimal library length, $L_{min}$, which is equal to the optimal embedding dimension ($E^*$), to the maximal library length, $L_{max}$, which is equal to the length of the entire time series. We used simplex projection[44] to identify the optimal $E^*$ for reconstructing the attractor, with E ranging from 1 to 10 for each time series. To guarantee convergence, we applied the following two statistical criteria[40]: (1) whether ρ(L) monotonically increased with L according to one-sided Kendall's τ test; and (2) whether $\rho(L_{max})$ was larger than zero according to one-sided Student's t-test. Convergence requires that both Kendall's τ test and Student's t-test are significant ($p < 0.05$). Detailed algorithm of CCM is in Supplementary Materials of Sugihara et al.[12].

We used CCM to test whether causal effects of age diversity, abundance, and environmental variables on population spatial variability existed. Because causality may occur with a lagged response[45], we also tested causal effect of lagged variable, with lag ranging from zero to eight quarters. To mitigate potential bias due to randomness in CCM analysis, we performed CCM analysis 200 times for each variable and retained the best lag at which ρ was the highest and often (95% of times) passed the convergence test. All time series were normalized by substracting mean and dividing standard deviation prior to any statistical analysis. The significant long-term trend in time series was removed using simple linear regression prior to analysis (i.e., by substracting the fitted value if the regression coefficient was significant with $p < 0.05$). It should be noted that, although CCM detects causations between variables in a pairwise manner, usage of lagged coordinate embedding from a single time series in state-space reconstruction implicitly incorporates influences of other causal variables in the dynamical system. This is because information about the entire dynamical system is recorded in any single time series[17,43], even though these confounding variables are not explicitly specified in the embedding model.

**Estimation of the time-varying direction and influential strength of causal effects**. Although CCM can identify causality, it cannot quantify the direction and influential strength of causal effects between variables. However, since the state of the dynamical system contains information of all variables, influential strengths between variables can be characterized by investigating the evolution of the state. In dynamical systems, the next time step of a variable, say $X_i(t)$, can be written as

$$X_i(t+1) = F(X_i(t)), \quad (7)$$

where F(·) denotes dynamics on the attractor. In a small neighborhood of $X_i(t)$, F(·) can be locally characterized by the Jacobian of $X_i(t)$. That is, the next time step of $X_i(t)$ is the net local effect that each of the variables in the same dynamical system has on $X_i(t)$. Hence, the Jacobian elements (i.e., partial derivatives) define influential strengths between variables on the state. In dynamic systems, influential strengths vary through time as the state evolves along the attractor[15], which is known as the state-dependent property. To obtain the unknown Jacobian at each time step, an algorithm called S-map[13,46] is introduced to sequentially reconstruct the Jacobian in an empirical way. The essential idea of S-map is to estimate the Jacobian at each successive state via a locally weighted multivariate linear scheme that gives greater weights to vector points near the current state in the reconstructed state space. That is,

$$\hat{x}_i(t^*+1) = C_0 + \sum_{j=1}^{E} C_{ij} x_j(t^*). \quad (8)$$

where $C_{ij}$ is model coefficient and E is the embedding dimension of the state space. S-map is conducted at each time step and gives greater weights to vector points closer to the target vector point $\mathbf{x}(t^*)$ on the attractor, which is unlike common linear models that ignore the state-dependent property. Exact weights are a function of the Euclidian distance d between vector points and $\mathbf{x}(t^*)$ normalized by

average distance $\overline{d}$ given by

$$w(d) = \exp\left(-\frac{\theta d}{\overline{d}}\right), \tag{9}$$

where $\theta \geq 0$ is a nonlinearity parameter determining the extent to which the model relies on the region near the target vector point. A larger value of $\theta$ indicates that the model is more state dependent and acts more like a nonlinear dynamical system. Note that when $\theta = 0$, the S-map model reduces to a vector autoregression model[40]. In short, coefficient **C** in the S-map is the singular value decomposition solution to the linear equation

$$\mathbf{B} = \mathbf{A} \cdot \mathbf{C}, \tag{10}$$

where **B** is an $n$-dimensional vector of the predicted next time-step variable given by

$$B_k = w(d(\underline{x}(t_k), \underline{x}(t^*)))x_i(t_k + 1), \tag{11}$$

and **A** is an $n$ by $E$ matrix of weighted variables given by

$$A_{kj} = w(d(\underline{x}(t_k), \underline{x}(t^*)))x_j(t_k). \tag{12}$$

We applied the S-map algorithm to estimate the time-varying influential strengths of the CCM-determined causal variables on population spatial variability. Specifically, we firstly used population spatial variability and the CCM-determined causal variables as coordinates to reconstruct the original attractor governing the dynamical system based on the extended Takens' theorem[17]. To satisfy the embedding theory, number of embedded causal variables should be equal to the optimal embedding dimension ($E^*$) of the system ($E^*$ can be determined by simplex projection as mentioned above). Hence, in addition to the target variable (i.e., population spatial variability), we selected another $E^* - 1$ critical causal variables of the target variable determined in CCM analysis (Table 1) to reconstruct the attractor. If $E^*$ was larger than the total number of causal variables, the suboptimal $E$ was used. Although $E^*$ is the best embedding dimension to reconstruct the attractor, suboptimal $E$ is also a practical embedding dimension and can be used to describe essentially the behavior of the attractor[44]. Thereafter, we used S-map to estimate influential strengths of each embedded causal variable on population spatial variability at each time step[46]. The nonlinearity parameter $\theta$ was tuned from zero to eight to identify the best $\theta$ for each S-map model. Performance of S-map model can be measured by the correlation coefficient $\rho$ between predicted and observed data. We tested whether $\rho$ is significantly larger than zero using one-sided Student's $t$-test. Notice that through S-map analyses, intertwining effects among age diversity, abundance, and environmental variables were examined simultaneously.

We noted that the number of significant causal variables for population spatial variability might be greater than $E^* - 1$. Therefore, there could be more than one valid state-space reconstruction to represent the attractor. In the main text, we presented results of state-space reconstruction using top $E - 1$ causal variables (according to the rank of $\rho$ in CCM analysis). However, to ensure robustness of our conclusion regarding the direction of the causal effect of each variable, we further examined all other candidate S-map models, using various combinations of alternative CCM-determined causal variables as a sensitivity analysis (i.e., replacing causal variables with those that were less strong, but still significant in CCM analysis). We performed this additional analysis for saithe and Norway pout because their number of significant causal variables for population spatial variability exceeded $E^* - 1$ (see Table 1 for $E^*$ and significant causal variables for each species).

**Aggregation tendency**. Taylor's exponent $b$ is a common indicator for describing aggregation tendency of a species, and normally lies between one and two. Larger $b$ indicates stronger aggregation tendency (i.e., a population becomes more aggregated (with higher population variance) when abundance increases; $b = 1$ represents a random spatial distribution[5,23,33]. We calculated Taylor's exponent $b$ by fitting a linear regression on the quarterly (Q1 and Q3) log-transformed variance of CPUE ($V$) with log-transformed mean of CPUE ($M$) over subareas: $\log(V) = \log(a) + b \times \log(M)$. Subareas with consistently zero CPUE throughout the study period were removed prior to analysis to avoid including areas where a species never inhabited.

**Analyses involving fishing mortality**. To explore potential indirect causal effect of fishing on population spatial variability by affecting age structure and abundance, we collected data of fishing mortality. Because only yearly data were available for fishing mortality, all analyses involving fishing mortality were done on a yearly base. That is, we averaged data over quarters in each year to obtain yearly data for each time series prior to any analysis. We performed CCM analysis and tested lagged variables as described above, but with lag up to four due to limited time-series length. S-map was done similarly; however, to reduce variability arising from the short time series in the S-map model, we averaged influential strengths of each causal variable over all significant lagged terms.

**Computation**. All analyses were done with the statistical software R (version 3.5.0). CCM and S-map analyses were implemented using the R package rEDM (version 0.6.9) (https://cran.r-project.org/web/packages/rEDM/index.html).

**Reporting summary**. Further information on research design is available in the Nature Research Reporting Summary linked to this article.

## Data availability

All raw data that support the findings of this study are publicly available. Fish survey data are available at https://datras.ices.dk/Data_products/Download/Download_Data_public.aspx. SBT was downloaded at https://ocean.ices.dk/HydChem/HydChem.aspx?plot=yes. SST was accessed at https://www.esrl.noaa.gov/psd/data/gridded/data.cobe2.html. AMO data were downloaded at https://www.esrl.noaa.gov/psd/data/timeseries/AMO/. Fishing mortality data can be retrieved from http://standardgraphs.ices.dk/stockList.aspx by specifying species, regions, and year. Life style of study species was accessed at http://www.fishbase.org/search.php by specifying species name. Raw and compiled dataset have been made publicly available at the repository https://doi.org/10.5281/zenodo.3759382.

## Code availability

R-codes and documentation of all analytical procedures mentioned above have been made publicly available at GitHub. Specifically, codes for parsing all surveys in DATRAS are available at https://github.com/snakepowerpoint/compileDATRAS. Codes for analyzing population spatial variability with EDM are available at https://github.com/snakepowerpoint/SpatialVariability.

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

## Acknowledgements

We thank the International Bottom Trawl Survey Working Group (IBTSWG) and all researchers included in the IBTS for providing data. We benefited from discussion with Chun-Wei Chang, Jin Kao, and Hsiao-Hang Tao. Dr. John Kostelic helped in English editing. This work was supported by the National Center for Theoretical Sciences, Foundation for the Advancement of Outstanding Scholarship, and the Ministry of Science and Technology, Taiwan, under grant: MOST107-2313-B-002-043 and MOST107-2636-B-019-001.

## Author contributions

J.-Y.W. and C.-h.H. conceived the research idea. J.-Y.W. and T.-C.K. analyzed the data with assistance from C.-h.H., J.-Y.W., T.-C.K., and C.-h.H. wrote the manuscript.

## Competing interests

The authors declare no competing interests.

## Additional information

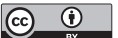

