## [Peer Review File · Nature Communications]

Reviewers' comments:

Reviewer #1 (Remarks to the Author):

Wang et al. examine the spatial variability of 9 commercially harvested fish species in the North Sea in relation to population age composition, abundance, and two environmental drivers (water temperature and AMO). The authors implement CCM and S-Map to determine causality of these variables. They find that for five of the 9 examined species there are significant causality between the examined variables and the population spatial variability. Age composition negatively affects spatial variance, in agreement with a-priori hypothesis. The effect of population abundance depends on the aggregative behavior of the population, as measured with the Taylor exponent, connecting the mean to the variance of a population. Results of environmental drivers are somewhat mixed, and related to the thermal niche and behavioral trait of the population.

Overall the study is sound, results are novel, and implications are examined. My concern is in the potential relevance of this study to the wide readership of Nature Communication. The processes examined in this study are well established. Many articles have examined age-dependent effects of environmental variability on fish distribution. Similarly, there is a long history of studies addressing effects of population abundance on spatial distribution in both terrestrial and marine systems. What is novel here are the methodologies (albeit hard to reproduce), and the fact that all of these effects are examined together.

I have also identified issues with the author's interpretation and implications of the results. Some of the claims are speculative or overreaching, given the results and intervening methodologies.

Detailed comments are provided below, in the order of importance.

L 79: Unless I get it wrong, these methods (CCM and S-map) are still correlative in nature. They do not isolate mechanisms (e.g., whether metabolic or foraging processes affect spatial distribution), but rather variables that are linked to the processes examined via non-linear and non-stationary dynamics. In fact, the performance of the S-map model is measured by Pearson correlation coefficient (ρ), which is a measure of statistical covariance among a set of variables. Therefore, I have troubles understanding the statement of L 78, stating that: 'These methods, in contrast to linear approaches, can distinguish causality from correlation by depicting underlying mechanisms of a dynamical system'

L 187: The assertion that the aggregation tendency explains the effect of abundance on spatial variance via the exponent 'b' seems somewhat circular to me. Would not the fact that b is > 2 in populations with negative abundance effects on spatial variance (i.e., cod) be implicit in the data? This is like saying that populations with spatial variance greater than the mean are aggregated in space. This result does not 'provide an explanation for previous contradictory observations', because we still do not know why cod aggregates instead of dispersing (like most species do) when their abundance increases.

L125: Some of the results implications are speculative and hard to understand. For example, only 2 out of 9 species examined showed a negative effect of age diversity on population spatial variability. Is this enough to state that 'To a large extent, this supported our first hypothesis that increasing age diversity reduces population spatial variability'? Similar consideration apply to abundance (3 out of 9 significant negative effects)

L 203: Given previous considerations on age-specific distribution and effect of age diversity on spatial variance, one would expect that the effect of environmental variability on spatial variance is interacted with population age structure. In other words, each age group has its own response to temperature, and the effect of temperature on the spatial structure of the entire population depends on the age composition of the population. Also, here and elsewhere the authors keep on referring to local populations, but there is no evidence that the stock examined are indeed composed by multiple local populations. These are likely different age specific aggregations (i.e., different cohorts) rather than subpopulations.

L 154: The bet-hedging strategy to alleviate harmful effects on local population does not hold when the spatial variability is due to different age structures (as opposed to different local populations).

L185: The text refers to 'species (implying plural) showing positive abundance effect when b is greater than 2'. However, of the 5 examined, only cod had a $b > 2$. So species should be singular ('the species').

L 215: The argument that spatially heterogeneous temperature reduces survival of larvae from spawning to nursery habitats needs to be further explained. What causes a reduction of larval connectivity between these two habitats? Also the argument that this processes inhibits connectivity between subpopulations does not hold. Individual that disperse from spawning to nursery habitats are not connecting different subpopulations.

L 307: I am having hard time following the narrative regarding CV calculations. I think that having equations with sub-indexes for each subarea, time, species, and age group would help. Same goes for abundance, age diversity and environmental variability.

L 327 Methods for convergent cross-mapping are hard to understand. I do not question the accuracy, but I do think that the methods can be better explained, maybe introducing less ambiguous terms, explaining jargon, and introducing equations. For example, what are library lengths?

Paragraph starting on L 169: From what is written in the main text, it is not clear to me how the Taylor exponent relates to increase/decrease of spatial variance at higher abundance. Would higher tendency to aggregate lead to higher spatial variability at low abundance? However, these issues are clarified in the methods. So I suggest moving some of the text, currently in the methods (specifically L 395-401) into the main text.

L 109: Sentence seems contradictory. Results were consistent with previous studies (therefore not new), but had hitherto not been systematically demonstrated?
Please elaborate.

L 220: What does 'weakening the strength of spatial synchrony' mean?

L192: remove instead

L 208: remove instead

L 209: Again, this is age specific

L 212: Remove 'also'

L 115: Clarify what is meant by 'coupled'. Correlated? Collinear?

Abstract L 37: Species (plural)? Only cod showed this tendency

Reviewer #2 (Remarks to the Author):

Review of: "Causal effects of age structure, abundance and environment on spatial variability of marine fishes", by Jheng-Yu Wang, Ting-Chun Kuo, Chih-Hao Hsieh

Submitted for consideration by Nature Communications

Review by Dr Coby Needle, Marine Scotland Science, Aberdeen (21 June 2019)

Summary

I found this to be an interesting paper, with a novel application of an existing methodology to the analysis of fisheries survey data, and it could be a useful addition to the extant literature. I cannot question the methods used or their implementation, but there are a number of issues with the selection of species, and the conclusions are not as robust as the text would imply. Therefore I would recommend acceptance after major revision.

Comments

1. My main concern with the paper lies with the selection of species. It is generally a good rule of thumb for a study like this to only consider those species for which the IBTS survey is used in the ICES stock assessment, as this indicates that the relevant experts consider the survey to be sufficiently representative of stock abundance and distribution. This is the case for cod, haddock, whiting, and Norway pout in this study – but not for herring, plaice, saithe, mackerel and sprat. Specifically: mackerel, herring and sprat are pelagic species for which a bottom-trawl survey such as IBTS is very unlikely to be appropriate, and which use acoustic survey data instead. Plaice is a benthic flatfish for which catchability in the IBTS survey is questionable: individuals are likely to go under the footrope of the gear, and for this reason the ICES assessment uses the beam-trawl survey index. Saithe is a northerly species, and the IBTS survey covers only the very southern fringes of its distribution. I would suggest that the authors consider this issue, and use the ICES data in DATRAS that is relevant to the species concerned.
2. The conclusions are not as robust as the text would imply. Figure 1 suggests that age diversity has a significantly negative effect on spatial variability for 2 out of the 4 species shown, which is thin ground on which to base the statement (line 245): "Declining age diversity elevated population spatial variability". The influence of abundance is clearly much stronger, so the conclusions there are appropriate. Temperature and the CV of temperature both increased spatial variability in 2 out of 3 significant cases, which again is not a strong basis for the conclusion (line 249): "Warming and spatially heterogeneous temperature enhanced population spatial variability". I would recommend that the conclusions be modified to reflect the outcomes of the analysis more directly.
3. I didn't find the brief comments on the impact of fishing mortality to be very convincing. The

strongest spatial signal in the North Sea cod stock, for example, has been a sharp decline in the southern and central areas and a more sustained population in the north. This could be due to environment, but equally it could be the result of spatially heterogeneous fishing pressure, and I think this study could provide a useful approach to explore which of these is more influential. Otherwise, the paper will always be open to the criticism that it has missed one of the key determinants of fish distribution.

4. The AMO acronym is used for the first time on line 108, but without a definition (that is left to the supplementary material).
5. It is not clear which year Figure 2 and Supplementary Figure 2 relate to – or is it all years? If it is the latter, that could cause problems as distribution has changed significantly for many of these stocks over time.
6. Line 178: I can see how the equation can be rewritten like this, but I can't see how either of those terms represents population spatial variability – perhaps this could be explained more clearly.
7. Line 207: it is wholly incorrect to state that mackerel has "limited movement under a changing environment". The Atlantic mackerel stock is one the most mobile in the world, and recent years have seen significant changes in migration paths due to environmental differences. Saithe are also a very mobile species, probably more so than cod.
8. The discussion on cod and the hypothesised aggregating tendency is not very convincing. The conclusion is based on an equation which I cannot see the relevance of (see point 6 above), and it also relies on the view that cod are significantly more mobile than the other species included here – which I don't believe is the case.
9. Line 233: the comment about "ambiguous causal effects" could apply to nearly all of the effects considered in this study, with the exception of abundance. In no other case is the outcome strong enough to warrant the full conclusions drawn in the paper, and I would encourage the authors to ensure that the text reflects this.
10. I do agree that population spatial structure is important to the management of these stocks, and hence I concur fully with the conclusion in line 263 onwards. I don't think the analyses are strong enough to support that conclusion yet, though.
11. Methods (Fish data): The IBTS database contains data on many more than 9 species. Also, did the authors ensure that the CPUE data was measured over consistent haul lengths (as these can vary considerably)?
12. Methods (Population spatial variability): The term "subarea" is not defined. Is it an ICES stat square, or a sampling area? If it is the latter, these are all of different sizes, and that might have an impact on the outcome.
13. Methods (Age structure...): Are CPUE data calculated by number or by weight? The source of the environmental data is also not noted here.
14. I found the link to the video (line 334) to be very useful indeed.
15. Acknowledgements: Which ICES working group is referred to here?
16. Figure 1: The additional key (bottom right) is not required and can be confusing. The labels should also be explained more clearly.
17. Figure 2: These plots suggest than the survey consists of one haul in the centre of each stat square, which is not the case. The bubbles should be shown at the relevant haul locations. Also, which year is this, or it is all years?
18. Supplementary Figure 2: I'm not sure this is very helpful – it is an extremely dense plot that requires a lot of work from the reader to understand. It's also not clear which year is referred to.

Reviewer #3 (Remarks to the Author):

The paper by Wang, Kuo & Hsieh is a novel and highly original analysis of causal drivers of spatial

variability of fisheries in the North Sea. The paper uses the powerful empirical dynamic modelling (EDM) approach that enables causal hypotheses to be tested using empirical data, in the absence of complex mechanistic model equations. The hypotheses tested were related to expectations of the influence of age truncation, abundance and climatic drivers on spatial variability of 9 commercial fish species in the North Sea. Both intensive fishing and climate change are known to have affected fish stocks, but past studies have either been correlational or rely on mechanistic models with detailed assumptions. Although plenty of work on abundance-occupancy relationships and density dependent habitat selection has been done to examine the apparent effects of fishing previous work doesn't specifically examine causal links affecting spatial variability as a response to fishing. Here, the authors show, using EDM, that there are clear interactions between each of the drivers and spatial variability that differ across species but that are mostly consistent with their hypotheses.

The icing on the cake for me was the final analysis of Taylor's Law exponents (from variance vs mean abundance relationship) that were used to explain counterintuitive findings in the causal relationships. More specifically, Atlantic cod had an exponent > 2 implying aggregative behaviour influenced the finding of reduced age diversity resulting in lower (not higher) spatial variability.

I have a couple of comments that relate to how the overriding hypotheses were tested. For age truncation and reduced abundance (hypotheses 1 and 2) these are supposed to be symptoms of fishing impacts. I wondered why the authors did not look at estimates of fishing mortality rates from stock assessments (publicly available from ICES) for these fish species to also determine whether there was higher spatial variability with increased levels of fishing mortality rates, as this would seem to more directly test their hypotheses, or at least complementary?

I also found the counterintuitive results for Atlantic cod very intriguing, especially since there is empirical evidence of density dependent habitat selection that shows spreading out at levels of high abundance (although this does not necessarily contradict the study's findings as the degree of expansion/aggregation can vary greatly across species). As shown the temperatures have also been changing over the course of the time series studied – and this was shown to negatively influence spatial variability (for that species). I wondered whether there are interactions with the changing temperatures, in abundance and age structure, size structure or other variables not included that might mask or cause "mirage" effects with these models? For example, the optimal temperature depends on size as well as age. Smaller cod occupy warmer waters than larger one of which there may be more habitat volume. Perhaps they are also limited in how much they can spread out once they are large, so it may not just be the aggregative behaviour but the distribution of the suitable environment combined with their thermal range/preference. I also noticed the Cohen paper on variance-mass allometry was mentioned. The sizes of cod have changed. How would these changes influence or interfere with the above expectations? Would this affect the expectations regarding spatial variability for smaller species vs larger species (Norway pout vs. cod)? I'm not suggesting analyses here but it would be interesting to hear about the potential of interactions such as these to be dealt with in your methods and how their potentially hidden presence might influence interpretation of the results?

I had a minor comment that perhaps the lack of fit for the pelagic species may be affected by detection in the survey – the IBTS demersal trawl is not well designed to catch these species, or were stock assessments used? This is a caveat and perhaps warrants clarifying text in the methods or in the interpretation of those results.

Another minor comment relates to the need for time series stationarity in EDM. These time series are not stationary, please state how that issue was dealt with.

The paper contains sufficient information for reproducibility. Although the methods described require expertise and skill all of the pertinent guides are given and the authors have kindly provided a github link where their analyses are indeed repeatable.

I enjoyed reading and reviewing this paper. In my opinion this paper will be extremely interesting to a very wide range of readers. It will be highly influential for ecologists, more applied fisheries researchers and is very policy relevant.

-Julia Blanchard

Referee #1:

Wang et al. examine the spatial variability of 9 commercially harvested fish species in the North Sea in relation to population age composition, abundance, and two environmental drivers (water temperature and AMO). The authors implement CCM and S-Map to determine causality of these variables. They find that for five of the 9 examined species there are significant causality between the examined variables and the population spatial variability. Age composition negatively affects spatial variance, in agreement with a-priori hypothesis. The effect of population abundance depends on the aggregative behavior of the population, as measured with the Taylor exponent, connecting the mean to the variance of a population. Results of environmental drivers are somewhat mixed, and related to the thermal niche and behavioral trait of the population.

Overall the study is sound, results are novel, and implications are examined. My concern is in the potential relevance of this study to the wide readership of Nature Communication. The processes examined in this study are well established. Many articles have examined age-dependent effects of environmental variability on fish distribution. Similarly, there is a long history of studies addressing effects of population abundance on spatial distribution in both terrestrial and marine systems. What is novel here are the methodologies (albeit hard to reproduce), and the fact that all of these effects are examined together.

I have also identified issues with the author's interpretation and implications of the results. Some of the claims are speculative or overreaching, given the results and intervening methodologies.

Detailed comments are provided below, in the order of importance.

Response:

We thank the reviewer for the positive comments. Although the reviewer mentioned that many articles had examined age, abundance and environmental effects on population spatial distribution, our study is different from them in several aspects. First, previous studies mostly focused on how spatial range or boundary may change in response to age composition, abundance or environmental conditions. For example, truncated age structure and diminished abundance will result in contracting spatial distribution. Warming temperatures can drive marine fishes to move northward or to deeper water. In contrast, our study focused on variability of spatial distribution, which is more related to the internal structure of spatial distribution and is critically

linked with stability. To the best of our knowledge, this is the first study to clearly demonstrate that population spatial variability is causally linked with age structure, abundance and environment.

Second, in contrast to previous studies using correlation-based methods to determine the causation between variables, in this study we used methodologies (CCM and S-map) that can distinguish causality. Instead of correlation, CCM and S-map leverage the concept of state-space reconstruction to understand causation between variables (by investigating reconstructed attractors).

Third, we considered all causal variables together, including age diversity, abundance and environment, to estimate their causal effects on population spatial variability. Such an analysis of multiple confounding variables is rare and difficult to conduct using conventional correlation analyses. We are also one of the first to determine that truncated age structure may have harmful effect on population spatial structure via enhancing variability, which can weaken stability and sustainability of populations.

We thank the reviewer's reminder regarding our interpretation and implications of results. We have rewritten most parts of the interpretation of our results to avoid over-speculation (e.g., P14: L339-347). All analyses can be reproduced following R-codes provided in <https://doi.org/10.5281/zenodo.3518702>. Thus, our methodology is fully reproducible.

Comment 1:

L 79: Unless I get it wrong, these methods (CCM and S-map) are still correlative in nature. They do not isolate mechanisms (e.g., whether metabolic or foraging processes affect spatial distribution), but rather variables that are linked to the processes examined via non-linear and non-stationary dynamics. In fact, the performance of the S-map model is measured by Pearson correlation coefficient (ρ), which is a measure of statistical covariance among a set of variables. Therefore, I have troubles understanding the statement of L 78, stating that: 'These methods, in contrast to linear approaches, can distinguish causality from correlation by depicting underlying mechanisms of a dynamical system'

Response 1:

We emphasize that mechanisms behind CCM and S-map are indeed different from correlation-based methods; thus, CCM and S-map can determine causality whereas correlations-based methods cannot. CCM and S-map are based on state-space

reconstruction. In dynamical system theory, the manifold governing a dynamical system can be reconstructed in an embedding space consisted of lagged coordinates from a single time series. The reconstructed manifold, called shadow manifold, is identical to the true manifold in a topological sense and thus can depict behavior of a dynamical system. If a time series, say $X(t)$, is affected by another time series, say $Y(t)$, the shadow manifold of $X(t)$ will record information of $Y(t)$. Therefore, one can use the shadow manifold of $X(t)$ to predict the shadow manifold of $Y(t)$. Based on this concept, CCM tests the correspondence between **two shadow manifolds** (not the two time series). If the correspondence is significant, then the causation between two time series is examined. As such, CCM can identify whether a variable causally affects a target variable. Also, when there are other causal variables affecting the target variable, the causal effect of the examined variable will still be detected because the causal information is recorded in the target variable and can be recovered using lagged coordinate embedding, even though other confounding variables are not explicitly specified in the embedding model.

The Pearson correlation coefficient (ρ) – as mentioned by the reviewer – is for estimating performance of the prediction. That is, it measures the correlation between predicted value and observed value (NOT correlation between variables). A higher ρ means that performance of CCM or S-map model is better, but not meaning that correlation between variables is stronger.

We have revised our Methods to explain these approaches in more detail.

Comment 2:

L 187: The assertion that the aggregation tendency explains the effect of abundance on spatial variance via the exponent ‘b’ seems somewhat circular to me. Would not the fact that b is > 2 in populations with negative abundance effects on spatial variance (i.e., cod) be implicit in the data? This is like saying that populations with spatial variance greater than the mean are aggregated in space. This result does not ‘provide an explanation for previous contradictory observations’, because we still do not know why cod aggregates instead of dispersing (like most species do) when their abundance increases.

Response 2:

Taylor’s exponent and the spatial variability (coefficient of variation (CV) of spatial abundance) describe different aspects of a population’s aggregation pattern. Spatial variability only indicates the aggregation at a certain time point, whereas Taylor’s

exponent represents the “potential of aggregation” of a population – how much it will aggregate when its abundance increases. Therefore, a population with a larger Taylor’s exponent may **not** be very aggregated at all times of observations (Kuo *et al.* 2016 *Ecology*). Thus, spatial variability and Taylor’s exponent have very different meanings of population dynamics and they are not circular.

After rewriting Taylor’s equation ($CV = a'M^{\frac{b}{2}-1}$), we hypothesize that species having b larger than 2 should exhibit positive interaction between abundance and population spatial variability. Namely, when abundance increases, spatial variability will also increase, because the exponent $\frac{b}{2} - 1$ term is positive. It is saying that species with a stronger aggregation tendency ($b > 2$) are more likely to aggregate when abundance increases. In contrast, if a species has a lower aggregation tendency ($b < 2$), it will become less aggregated when its abundance increases. Our results, though with limited number of species, were consistent with such a hypothesis and verified the viability of rewriting Taylor’s equation. By examining the relationship between Taylor’s equation and abundance effect on population spatial variability, we provided a theoretical explanation for the observed contradictory abundance effects. However, we agree that Taylor’s equation does not provide further information on why cod aggregates when abundance increases, because Taylor’s exponent represents integrated outcome in which many processes are involved. Examining the underlying mechanism and the corresponding biological meaning of Taylor’s equation is beyond the scope of this study. Nevertheless, our deduction and results provided an insight that Taylor’s exponent can be an indicator suggesting how population spatial variability may respond to changes in abundance.

We revised the Methods to explain the computation and meanings of Taylor’s equation. A more detailed explanation of the spatial variability (CV) versus Taylor’s exponent has been carefully discussed in previous studies; thus, we feel that we do not need to provide details again in this manuscript.

Comment 3:

L125: Some of the results implications are speculative and hard to understand. For example, only 2 out of 9 species examined showed a negative effect of age diversity on population spatial variability. Is this enough to state that ‘To a large extent, this supported our first hypothesis that increasing age diversity reduces population spatial variability’? Similar consideration apply to abundance (3 out of 9 significant negative

effects)

Response 3:

We agree with the reviewer that our claims might be too assertive. We revised our interpretation of the results in a more conservative manner throughout the manuscript (e.g, P5-6: L124-134).

Comment 4:

L 203: Given previous considerations on age-specific distribution and effect of age diversity on spatial variance, one would expect that the effect of environmental variability on spatial variance is interacted with population age structure. In other words, each age group has its own response to temperature, and the effect of temperature on the spatial structure of the entire population depends on the age composition of the population. Also, here and elsewhere the authors keep on referring to local populations, but there is no evidence that the stock examined are indeed composed by multiple local populations. These are likely different age specific aggregations (i.e., different cohorts) rather than subpopulations.

Response 4:

Although we agree that temperature may indirectly affect population spatial variability through age structure for some species, temperature can also directly affect population spatial variability. To examine the confounding effect between age diversity and other variables, we conducted a new analysis in this revision that used CCM and S-map to examine factors affecting age diversity (P11-12: L276-288). We determined that for Atlantic mackerel, temperature had a significant positive effect on spatial variability (Table 1) and significant negative effect on age diversity (Supplementary Table 6). Therefore, we inferred that warming temperature may increase population spatial variability through lowering age diversity for mackerel. However, we also determined that without significant effect on age diversity, temperature had strong causal effect on spatial variability of Atlantic cod and plaice (Table 1 and Supplementary Table 4). Such findings indicate that a direct effect of temperature on population spatial variability exists for those species. The direct effect happens when the thermal condition of a certain location becomes unfavourable, local reduction or depletion may occur for all individuals in that location, irrespective of age class (Radlinski *et al.* 2013 *ICES J. Mar. Sci.*). This will result in a heterogeneous population spatial structure.

Regarding the “local population” or “subpopulation” referred to in the manuscript, we

meant to indicate that different groups of fishes that live in different locations, but not specifically suggesting existence of metapopulations. We have revised sentences in the manuscript where using such a term could cause confusion.

Comment 5:

L 154: The bet-hedging strategy to alleviate harmful effects on local population does not hold when the spatial variability is due to different age structures (as opposed to different local populations).

Response 5:

We disagree that the bet-hedging strategy will not hold when spatial variability is affected by age structure. First, spatial distribution of a population is not only affected by age structure (as apparent in our results (Table 1) and other studies (e.g. Nye *et al.* 2009 *Mar. Ecol. Prog. Ser.* and Radlinski *et al.* 2013 *ICES J. Mar. Sci.*)). Even if the age-specific distribution is the greatest driver of spatial heterogeneity, distributions of different cohorts overlap. A more homogeneous distribution has a higher chance to preserve the population when some habitats are no longer suitable. Even in the extreme case – each type of habitats is only occupied by a certain age class – the wider types of habitats occupied infer that other age classes can continuously reproduce when a certain age class (habitat) is eliminated. Bet-hedging strategy has a role in alleviating harmful effects from environmental disturbances for population sustainability, independent of whether or not age structure is the **only** factor of spatial variability.

Comment 6:

L185: The text refers to ‘species (implying plural) showing positive abundance effect when b is greater than 2’. However, of the 5 examined, only cod had a $b > 2$. So species should be singular (‘the species’).

Response 6:

We revised the wording accordingly (**P8: L187-190**).

Comment 7:

L 215: The argument that spatially heterogeneous temperature reduces survival of larvae from spawning to nursery habitats needs to be further explained. What causes a reduction of larval connectivity between these two habitats? Also the argument that this processes inhibits connectivity between subpopulations does not hold. Individual that disperse from spawning to nursery habitats are not connecting different

subpopulations.

Response 7:

We thank the reviewer for the comment. We had a logic flaw in the original argument and therefore revised the sentence.

(P9: L215-217) In addition, spatially heterogeneous temperatures can reduce survival of larvae and juvenile fish when they move from a spawning ground to a nursery ground²⁷, leading to fragmented spatial distribution of the population.

The spatial heterogeneous temperature will reduce survival rate of larvae when fishes move from habitat to another, because they may have a higher chance to encounter unsuitable environmental conditions if the temperature varies considerably across space. We understand that dispersal from spawning to nursery habitats are not connecting subpopulations; therefore, we revised the sentence.

Comment 8:

L 307: I am having hard time following the narrative regarding CV calculations. I think that having equations with sub-indexes for each subarea, time, species, and age group would help. Same goes for abundance, age diversity and environmental variability.

Response 8:

We have added the relevant equations as suggested (e.g. P15-16: L381,388,396,399).

Comment 9:

L 327 Methods for convergent cross-mapping are hard to understand. I do not question the accuracy, but I do think that the methods can be better explained, maybe introducing less ambiguous terms, explaining jargon, and introducing equations. For example, what are library lengths?

Response 9:

We revised the Introduction and Methods to explain the methods more clearly and added equations to improve understanding (e.g. P17: L417; P19-20: L465-498).

Comment 10:

Paragraph starting on L 169: From what is written in the main text, it is not clear to me how the Taylor exponent relates to increase/decrease of spatial variance at higher abundance. Would higher tendency to aggregate lead to higher spatial variability at

low abundance? However, these issues are clarified in the methods. So I suggest moving some of the text, currently in the methods (specifically L 395-401) into the main text.

Response 10:

As Response 2, Taylor's equation describes the relation between population spatial variability and abundance and the direction of the relation is determined by Taylor's exponent b . If a species has higher tendency to aggregate ($b > 2$), it will have higher spatial variability at higher abundance. We moved some sentences from the Methods to the main text to clarify such a relation, as suggested by the reviewer (**P7-8: L174-186**). We thank the reviewer for this suggestion.

Comment 11:

L 109: Sentence seems contradictory. Results were consistent with previous studies (therefore not new), but had hitherto not been systematically demonstrated? Please elaborate.

Response 11:

Based on our results, age diversity, abundance and environment all have causal effects on population spatial variability, consistent with results from many previous studies that have examined the correlation between spatial distribution and each or some of the variables. However, previous studies either tested causal effect by each factor or inferred causality based on correlation, which can give misleading results when there are omitted variables or when the system is nonlinear.

Comment 12:

L 220: What does 'weakening the strength of spatial synchrony' mean?

Response 12:

In the original sentence, "weakening the strength of spatial synchrony between abundance and temperature", we referred to the finding that spatial variability of abundance changes in the same direction with spatial variability of temperature. We modified the sentence to be easier to understand:

(P9: L219-221) *We inferred that their stronger movement responding to changes in temperature¹⁰ may enable them to inhabit suitable habitats more homogeneously; therefore, spatial variability of abundance was less affected by that of temperature.*

Comment 13:

L192: remove instead

Response 13:

We edited the sentence accordingly.

Comment 14:

L 208: remove instead

Response 14:

We edited the sentence accordingly.

Comment 15:

L 209: Again, this is age specific

Response 15:

As Response 4, temperature may have direct or indirect causal effect on population spatial variability, depending on species.

Comment 16:

L 212: Remove 'also'

Response 16:

We edited the sentence accordingly.

Comment 17:

L 115: Clarify what is meant by 'coupled'. Correlated? Collinear?

Response 17:

"Coupled" means variables are causally linked. As is apparent in CCM results (Table 1), population spatial variability is causally affected by nearly all examined variables, with a various extent in casual effects (ρ is within [0, 1]).

Comment 18:

Abstract L 37: Species (plural)? Only cod showed this tendency

Response 18:

We updated the results and revised the Abstract.

Referee #2:

Review of: “Causal effects of age structure, abundance and environment on spatial variability of marine fishes”, by Jheng-Yu Wang, Ting-Chun Kuo, Chih-Hao Hsieh

Submitted for consideration by Nature Communications

Review by Dr Coby Needle, Marine Scotland Science, Aberdeen (21 June 2019)

Summary

I found this to be an interesting paper, with a novel application of an existing methodology to the analysis of fisheries survey data, and it could be a useful addition to the extant literature. I cannot question the methods used or their implementation, but there are a number of issues with the selection of species, and the conclusions are not as robust as the text would imply. Therefore I would recommend acceptance after major revision.

Response:

Thank you for your compliments. We would like to clarify issues with the selection of species below. First, for raw data, we actually did not select species. Our study requires data containing spatial CPUEs for each age class over time. Such data were collected by IBTS for only nine standard species. We have clarified this concern in Methods (**P15: L363-367**). Second, for S-map analysis, we presented only the species which state-space was successfully reconstructed with the CCM-determined causal variables (having sufficient causal variables and the reconstructed manifold was significant) in the main text. Results of other species are in Supplementary Table 2.

We also revised our writing to ensure that our interpretation reflected results appropriately.

Comment 1:

My main concern with the paper lies with the selection of species. It is generally a good rule of thumb for a study like this to only consider those species for which the IBTS survey is used in the ICES stock assessment, as this indicates that the relevant experts consider the survey to be sufficiently representative of stock abundance and distribution. This is the case for cod, haddock, whiting, and Norway pout in this study – but not for herring, plaice, saithe, mackerel and sprat. Specifically: mackerel, herring and sprat are pelagic species for which a bottom-trawl survey such as IBTS is very unlikely to be appropriate, and which use acoustic survey data instead. Plaice is a

benthic flatfish for which catchability in the IBTS survey is questionable: individuals are likely to go under the footrope of the gear, and for this reason the ICES assessment uses the beam-trawl survey index. Saithe is a northerly species, and the IBTS survey covers only the very southern fringes of its distribution. I would suggest that the authors consider this issue, and use the ICES data in DATRAS that is relevant to the species concerned.

Response 1:

Our data (IBTS) are from DATRAS. We are aware that IBTS is designed for demersal stocks and therefore these data may not be the best index for pelagic species. However, to our knowledge, data used in this study were the best available data for investigating factors of spatial variability for fishes in the North Sea. Our analyses require age-specific spatial CPUE data and the time series need to be long enough for empirical dynamical modelling (EDM). Data containing only temporal information are not suitable for investigating spatial variability; unfortunately, few datasets meet such requirements. In addition, bottom trawl data may still be indicative for pelagic species, with the assumption that the fraction of catching these species by bottom trawl remains constant (Marquez *et al.* 2009 *Ecology Letters*). We added a paragraph (P12: L296-305) to clarify this caveat of using bottom trawl data for pelagic species and stated our assumption clearly. We also explicitly stated our caveats in the Methods.

We understand that regions defined by IBTS may not cover complete living areas for any of the stocks. However, considering the long time scale and large geographical range that IBTS covers, data from IBTS still have statistical relevance and provide many insights. We are confident that our results based on IBTS provided novel insights and are worthy of investigation.

Comment 2:

The conclusions are not as robust as the text would imply. Figure 1 suggests that age diversity has a significantly negative effect on spatial variability for 2 out of the 4 species shown, which is thin ground on which to base the statement (line 245):

“Declining age diversity elevated population spatial variability”. The influence of abundance is clearly much stronger, so the conclusions there are appropriate. Temperature and the CV of temperature both increased spatial variability in 2 out of 3 significant cases, which again is not a strong basis for the conclusion (line 249): “Warming and spatially heterogeneous temperature enhanced population spatial variability”. I would recommend that the conclusions be modified to reflect the

outcomes of the analysis more directly.

Response 2:

We modified the conclusions to appropriately reflect outcomes of analyses:

(P14: L342-347) Although with limited sample size, we determined that declining age diversity elevated population spatial variability, which had long been hypothesized^{1,4,5} but hitherto apparently never quantified. Reduced abundance either increased or decreased population spatial variability, probably depending on the aggregation tendency of a species. Warming and spatially heterogeneous temperatures may enhance population spatial variability, highlighting potential hazards to marine fishes under current global-warming scenarios.

Comment 3:

I didn't find the brief comments on the impact of fishing mortality to be very convincing. The strongest spatial signal in the North Sea cod stock, for example, has been a sharp decline in the southern and central areas and a more sustained population in the north. This could be due to environment, but equally it could be the result of spatially heterogeneous fishing pressure, and I think this study could provide a useful approach to explore which of these is more influential. Otherwise, the paper will always be open to the criticism that it has missed one of the key determinants of fish distribution.

Response 3:

The question that the reviewer proposed is very interesting and worth investigating. However, we do not have sufficient data to estimate region-wise fishing mortalities, and it is beyond the scope of this study. Despite this, we have added a new analysis regarding the fishing effect on age structure, abundance and population spatial structure. We determined that fishing had negative effects on age diversity and abundance for some species (Supplementary Table 6 and 7), given the short length of available time series. Therefore, we inferred that fishing was likely to affect population spatial structure through age structure and abundance. In addition, we determined fishing can directly affect population spatial variability (Supplementary Table 8), but the corresponding quantitative causal effects were difficult to estimate due to limited time series (Supplementary Table 9). Nonetheless, fishing indeed had causal effect on population spatial structure, irrespective of the direct or indirect pathway.

According to our results, both fishing and environment can affect population spatial

structure. It may not be possible to determine which one is more influential than the other when considering the potentially indirect causal effects arising from them. Actually, many of the examined variables were causally linked, as can be seen in our additional analyses on confounding effects among variables (P11-12: L276-288).

Comment 4:

The AMO acronym is used for the first time on line 108, but without a definition (that is left to the supplementary material).

Response 4:

We corrected this.

Comment 5:

It is not clear which year Figure 2 and Supplementary Figure 2 relate to – or is it all years? If it is the latter, that could cause problems as distribution has changed significantly for many of these stocks over time.

Response 5:

Figure 2 and Supplementary Figure 2 display average CPUE over years for each age class on each subarea. We are aware that the spatial distribution of each age class changes over time. However, we are not focusing on the changing spatial distribution of each age class. Here, our objective is to indicate that each age class has its own preferred habitat.

Comment 6:

Line 178: I can see how the equation can be rewritten like this, but I can't see how either of those terms represents population spatial variability – perhaps this could be explained more clearly.

Response 6:

We define the population spatial variability as the correlation of coefficient (CV) of the spatial abundance (CPUE). We thank the reviewer for the reminder and we made this point clearer in the main text (P4: L83 and P7: L174-179).

Comment 7:

Line 207: it is wholly incorrect to state that mackerel has “limited movement under a changing environment”. The Atlantic mackerel stock is one the most mobile in the world, and recent years have seen significant changes in migration paths due to

environmental differences. Saithe are also a very mobile species, probably more so than cod.

Response 7:

We apologize for the confusion. We meant to link the observation in our study to what was observed in Perry *et al.* 2005 *Science*; in that study, cod had significant spatial shifts with a warming environment, but saithe did not. Atlantic mackerel was not examined in that study. We speculate that the observed higher spatial variability with increasing temperature and temperature variability may be because the species do not respond to changes in temperature by moving. We revised the text accordingly:

(P8-9: L203-209) *Increasing population spatial variability with warming temperatures was observed in plaice and Atlantic mackerel (Table 2), which supported our third hypothesis. The resulting heterogeneous spatial distribution may have been due to a local reduction or a local extinction of populations that were not adapted to warming temperatures. This phenomenon is especially apparent for species that move less in a changing environment such as plaice¹⁰. On the contrary, a warming temperatures may decrease spatial variability of Atlantic cod, probably due to their relatively significant migratory ability in response to changing temperatures^{10,26}.*

Comment 8:

The discussion on cod and the hypothesised aggregating tendency is not very convincing. The conclusion is based on an equation which I cannot see the relevance of (see point 6 above), and it also relies on the view that cod are significantly more mobile than the other species included here – which I don't believe is the case.

Response 8:

We are not using species-specific mobile ability to explain the abundance effect on population spatial variability. Instead, according to the transformed Taylor's equation, we suggest that cod will become more aggregate when abundance increases because its aggregating tendency is higher (with $b > 2$). In contrast, the negative response of spatial variability to abundance observed in Atlantic mackerel may have been due to its lower aggregation tendency ($b < 2$). We infer that the abundance effect can be explained by Taylor's power law alone and is not related to species-specific mobile ability. We only leverage the species-specific mobile responses to the changing environment to explain results of environmental effects on the population spatial variability. We revised sentences accordingly.

Comment 9:

Line 233: the comment about “ambiguous causal effects” could apply to nearly all of the effects considered in this study, with the exception of abundance. In no other case is the outcome strong enough to warrant the full conclusions drawn in the paper, and I would encourage the authors to ensure that the text reflects this.

Response 9:

We revised the manuscript to make the conclusion more consistent with the results.

Comment 10:

I do agree that population spatial structure is important to the management of these stocks, and hence I concur fully with the conclusion in line 263 onwards. I don't think the analyses are strong enough to support that conclusion yet, though.

Response 10:

We agree that our S-map results may not be strong enough, due to a limited number of species and time-series length. However, our CCM results indicated significant causal effects of age structure, abundance and environmental variables on the spatial variability for almost all species examined. We believe that such results provide evidence that careful attention to these factors in fisheries management is needed.

Comment 11:

Methods (Fish data): The IBTS database contains data on many more than 9 species. Also, did the authors ensure that the CPUE data was measured over consistent haul lengths (as these can vary considerably)?

Response 11:

IBTS indeed contains data on many more than nine species. However, our study requires age-specific spatial data for each species over time. Data containing CPUE for each age class and each subarea are only found for nine species (ICES groups only do the age-specific spatial survey for nine standard species). Therefore, we could only use these nine species in this study. We clarified this concern in Methods (**P15: L363-367**).

The dataset we used was “CPUE per length per subarea” from IBTS, but not “CPUE per length per haul” (which is also available from IBTS). Thus, we do not have the issue of the discrepancy over haul length.

Comment 12:

Methods (Population spatial variability): The term “subarea” is not defined. Is it an ICES stat square, or a sampling area? If it is the latter, these are all of different sizes, and that might have an impact on the outcome.

Response 12:

“Subarea” is defined as the ICES stat square (they called the survey grid) in this study. Each subarea has equal size within 5°W-13°E and 49.5°N-61.5°N with the resolution 0.5°x1°, respectively. Therefore, there should be no issue with the irregular size of subarea. We added a sentence in **P15: L378** to clarify this definition:

Each subarea was defined as the ICES stat square (i.e., survey grid) and had equal size as 0.5°x1°.

Comment 13:

Methods (Age structure...): Are CPUE data calculated by number or by weight? The source of the environmental data is also not noted here.

Response 13:

CPUE data are based on the number of individuals. We do not have access to any weight-based CPUE data. We added a sentence in **P15: L367-369** to clarify this:

Catch data were calculated based on number of individuals, as no biomass data were available at the time of data retrieval.

The source of the environmental data is noted in Data Availability.

Comment 14:

I found the link to the video (line 334) to be very useful indeed.

Response 14:

Thank you.

Comment 15:

Acknowledgements: Which ICES working group is referred to here?

Response 15:

It referred to the International Bottom Trawl Survey Working Group (IBTSWG). We made this clearer in Acknowledgements.

Comment 16:

Figure 1: The additional key (bottom right) is not required and can be confusing. The labels should also be explained more clearly.

Response 16:

We modified Figure 1 and explained labels.

Comment 17:

Figure 2: These plots suggest that the survey consists of one haul in the centre of each stat square, which is not the case. The bubbles should be shown at the relevant haul locations. Also, which year is this, or it is all years?

Response 17:

As Response 11, the dataset we used was “CPUE per length per subarea” from IBTS, but not “CPUE per length per haul”. Therefore, each point in the figure indicated magnitude of CPUE and was positioned in the centre of each subarea. Figure 2 is the average CPUE over years for each age class on each subarea (please see Response 5 for more details).

Comment 18:

Supplementary Figure 2: I’m not sure this is very helpful – it is an extremely dense plot that requires a lot of work from the reader to understand. It’s also not clear which year is referred to.

Response 18:

Supplementary Figure 2 is the average CPUE over years for each age class on each subarea. We clarified this in the legend and simplified the figure.

Referee #3:

The paper by Wang, Kuo & Hsieh is a novel and highly original analysis of causal drivers of spatial variability of fisheries in the North Sea. The paper uses the powerful empirical dynamic modelling (EDM) approach that enables causal hypotheses to be tested using empirical data, in the absence of complex mechanistic model equations. The hypotheses tested were related to expectations of the influence of age truncation, abundance and climatic drivers on spatial variability of 9 commercial fish species in the North Sea. Both intensive fishing and climate change are known to have affected fish stocks, but past studies have either been correlational or rely on mechanistic models with detailed assumptions. Although plenty of work on abundance-occupancy relationships and density dependent habitat selection has been done to examine the apparent effects of fishing previous work doesn't specifically examine causal links affecting spatial variability as a response to fishing. Here, the authors show, using EDM, that there are clear interactions between each of the drivers and spatial variability that differ across species but that are mostly consistent with their hypotheses.

The icing on the cake for me was the final analysis of Taylor's Law exponents (from variance vs mean abundance relationship) that were used to explain counterintuitive findings in the causal relationships. More specifically, Atlantic cod had an exponent > 2 implying aggregative behaviour influenced the finding of reduced age diversity resulting in lower (not higher) spatial variability.

Response:

We thank the reviewer for the positive comments and thorough summary.

Comment 1:

I have a couple of comments that relate to how the overriding hypotheses were tested. For age truncation and reduced abundance (hypotheses 1 and 2) these are supposed to be symptoms of fishing impacts. I wondered why the authors did not look at estimates of fishing mortality rates from stock assessments (publicly available from ICES) for these fish species to also determine whether there was higher spatial variability with increased levels of fishing mortality rates, as this would seem to more directly test their hypotheses, or at least complementary?

Response 1:

We conducted additional analysis to investigate whether fishing mortality rates affect age structure, abundance and spatial variability, as the reviewer suggested (P10:

L235-240; P11: L266-274). We determined that fishing tended to negatively affect age diversity and abundance (Supplementary Table 6 and 7), consistent with previous observations on age truncation and abundance reduction arising from fishing. We also identified a significant causal effect of fishing on spatial variability (Supplementary Table 8); however, due to the limited time series, we were not able to successfully reconstruct the manifold and estimate quantitatively influential strengths of fishing effect on spatial variability for most species (Supplementary Table 9).

We acknowledge that the short time series of fishing mortality data may induce biases in this analysis. Since sampling frequency of fishing mortality rates is yearly but the other IBTS data are quarterly, combining these two datasets decreases the time-series length to 25 (yearly data from 1991 to 2015). Such time series may be too short for empirical dynamical modelling (EDM) to reconstruct attractor and estimate interaction strengths. This is because EDM (including simplex projection, convergence cross mapping (CCM) and S-map) is based on the concept of state-space reconstruction. If there are insufficient data, the reconstructed attractor will be incomplete and only reveal some parts of the ground-truth attractor. Meanwhile, averaging quarterly IBTS data to merge with yearly fishing mortality rates may lose much hidden information in IBTS data. Nevertheless, we have presented and discussed those results.

Comment 2:

I also found the counterintuitive results for Atlantic cod very intriguing, especially since there is empirical evidence of density dependent habitat selection that shows spreading out at levels of high abundance (although this does not necessarily contradict the study's findings as the degree of expansion/aggregation can vary greatly across species). As shown the temperatures have also been changing over the course of the time series studied – and this was shown to negatively influence spatial variability (for that species). I wondered whether there are interactions with the changing temperatures, in abundance and age structure, size structure or other variables not included that might mask or cause “mirage” effects with these models? For example, the optimal temperature depends on size as well as age. Smaller cod occupy warmer waters than larger one of which there may be more habitat volume. Perhaps they are also limited in how much they can spread out once they are large, so it may not just be the aggregative behaviour but the distribution of the suitable environment combined with their thermal range/preference. I also noticed the Cohen paper on variance-mass allometry was mentioned. The sizes of cod have changed. How would these changes influence or interfere with the above expectations? Would

this affect the expectations regarding spatial variability for smaller species vs larger species (Norway pout vs. cod)? I'm not suggesting analyses here but it would be interesting to hear about the potential of interactions such as these to be dealt with in your methods and how their potentially hidden presence might influence interpretation of the results?

Response 2:

We agree that these variables may actually intertwine with each other. Therefore, we conducted a new analysis to examine variables affecting age structure and abundance (P11-12: L276-288). We determined that both fishing and temperature influenced age structure as well as abundance, whereas temperature, together with age structure and abundance, also contemporarily affected population spatial variability

We agree with your points that abundance effect on population spatial variability may not just depend on aggregative behavior. For example, we determined that changes in abundance can influence age diversity of Atlantic mackerel (Supplementary Table 6), which in turn affected population spatial variability (Table 2). Therefore, abundance may indirectly affect population spatial variability via changes in demographic structure induced by varying levels of abundance. The underlying mechanisms behind abundance effect can be complex. The variance-mass allometry is also a possible mechanism for changes in spatial variability, as you suggested. Although we did not include the mean population size into the analyses in this study, we added sentences in P12: L289-293 to suggest this for future exploration.

Comment 3:

I had a minor comment that perhaps the lack of fit for the pelagic species may be affected by detection in the survey – the IBTS demersal trawl is not well designed to catch these species, or were stock assessments used? This is a caveat and perhaps warrants clarifying text in the methods or in the interpretation of those results.

Response 3:

We acknowledge that IBTS was not designed for catching pelagic species. However, our topic, to analyze age structure, abundance and environmental effect on population spatial variability, requires spatial and temporal data of each age class for each species. In the North Sea, IBTS is the only dataset that fits this setting. Certainly, IBTS is not an optimal dataset for pelagic species; however, some patterns still exist in this dataset, as is evident in our results. In addition, bottom trawl data may still be indicative for pelagic species, with the assumption that the fraction of catching these species by

bottom trawl remains constant (Marquez *et al.* 2009 *Ecology Letters*). We added a paragraph (P12: L296-305) to clarify this caveat of using bottom trawl data for pelagic species and stated our assumption clearly.

Comment 4:

Another minor comment relates to the need for time series stationarity in EDM. These time series are not stationary, please state how that issue was dealt with.

Response 4:

We thank the reviewer for this reminder. We re-did all analyses with stationary time series. For stationarity, we used simple linear regression to remove the long-term trend in time series if it exists. We updated our results in the current manuscript.

Most of the examined variables still had a significant causal effect on population spatial variability for all species, as suggested by CCM (Table 1). For the quantitative causal effect measured by S-map, four of the nine study species had significant results (Table 2). Age diversity still had a negative causal effect on population spatial variability for species with significant age diversity effect. Abundance effect was still consistent with our hypothesis about Taylor's exponent, though only two species had significant abundance effect here. AMO effect remained unclear. Temperature and variability of temperature were prone to positively affecting population spatial variability for two species respectively; however, cod still showed the opposite results.

Overall, results were qualitatively similar to the previous one, albeit with one less species had significant S-map results. This was probably because the time series loss some information when detrending, especially when time series were short (as in this study). We really appreciate that the reviewer noted this methodological issue of time series stationarity in our analyses.

Comment 5:

The paper contains sufficient information for reproducibility. Although the methods described require expertise and skill all of the pertinent guides are given and the authors have kindly provided a github link where their analyses are indeed repeatable.

Response 5:

Thank you.

Comment 6:

I enjoyed reading and reviewing this paper. In my opinion this paper will be extremely interesting to a very wide range of readers. It will be highly influential for ecologists, more applied fisheries researchers and is very policy relevant.

Response 6:

Thank you.

Reviewers' comments:

Reviewer #1 (Remarks to the Author):

The authors have done a great job in addressing reviewers concerns, and clarifying questions. After re-reading the manuscript, my only suggestion is that of adding figures pointing to empirical evidence of the results. For example, Fig. 2: instead of showing distribution by age for *Gadus morhua*, I suggest plotting two population distribution patterns, one during a period characterized by low age diversity, and another characterized by high age diversity. Similar plots can be done for other variables (temperature and abundance) and species. These empirical figures should visually corroborate the findings from the analysis.

Reviewer #2 (Remarks to the Author):

Many thanks to the authors for their careful and considered responses to the points I and the other reviewers raised. I have the following additional responses in turn:

Response to Summary: The IBTS data in DATRAS is actually much more extensive than the authors have realised. They have used only species for which CPUE per stat square per age data are presented, but in fact the Exchange data format is extremely extensive and I am confident that many more species would have been available for the analysis had the authors known how to access and analyse Exchange data. Various tools are available as R libraries to help with this, and ICES staff are very willing to help with this (I could provide assistance if required as well).

Response 1: I'm afraid I don't accept this explanation, and I return to my concern that the authors are using data for stocks for which they are not suited. An assumption of constant bottom fraction does not hold for mackerel, which are very pelagic and whose position in the water column is certainly not fixed. For herring and sprat, they could have used the acoustic data which is also available (albeit in a beta form) through the ICES website. For plaice, they should use the beam trawl survey (BTS) data which covers a large proportion of the plaice stock, uses gear designed for plaice, and is available from DATRAS. Saithe in the North Sea will always be difficult to model using North Sea surveys, as only a small fraction of the stock actually is found in the North Sea. I am concerned about the validity of the authors' very clever modelling for these species for this reason.

Response 2: The point here is that this is the case for just 2 out of 4 significant cases, which does not justify the statement made about the effect of age diversity.

Response 3: I'm pleased to see this new section, I think it strengthens the paper a good deal.

Response 5: But if spatial distribution has changed over time, wouldn't that affect your conclusions about preferred habitat?

Response 10: Again, I think this is overplaying it - the results were significant and positive for much less than "almost all" species examined.

Response 11: See my comment above on the Exchange data format, which I think would be much more useful for these analyses (and cover many more relevant species).

Dr Coby Needle

Reviewer #3 (Remarks to the Author):

The authors have done a thorough job of addressing these comments. I have no other issues with the manuscript.

Reviewer #1:

The authors have done a great job in addressing reviewers concerns, and clarifying questions. After re-reading the manuscript, my only suggestion is that of adding figures pointing to empirical evidence of the results. For example, Fig. 2: instead of showing distribution by age for *Gadus morhua*, I suggest plotting two population distribution patterns, one during a period characterized by low age diversity, and another characterized by high age diversity. Similar plots can be done for other variables (temperature and abundance) and species. These empirical figures should visually corroborate the findings from the analysis.

Response:

Thank you for the constructive suggestions. We have added the figure (as new Figure 2 in revision) as suggested. As can be seen from Figure 2, populations tended to distribute more evenly in space when their age diversity was higher. In response to your comment, we also made similar plots for abundance and environmental effects (Supplementary Figures 3-8). Nevertheless, one should keep in mind that the causal effects of these factors on spatial variability are context-dependent; that is, there is no simple association between cause and target variables. Thus, these figures are just for visualization purposes and cannot be used as statistical evidence.

Reviewer #2:

Many thanks to the authors for their careful and considered responses to the points I and the other reviewers raised. I have the following additional responses in turn:

Response to Summary: The IBTS data in DATRAS is actually much more extensive than the authors have realised. They have used only species for which CPUE per stat square per age data are presented, but in fact the Exchange data format is extremely extensive and I am confident that many more species would have been available for the analysis had the authors known how to access and analyse Exchange data. Various tools are available as R libraries to help with this, and ICES staff are very willing to help with this (I could provide assistance if required as well).

Response to summary:

Thank you for providing the information about DATRAS. We especially appreciate your comments and help through email exchanges that made a careful examination of DATRAS possible. We have followed the suggestions to examine the Exchange data of DATRAS to explore if we can include more species/surveys into our analyses. However, the purpose and method (state-space reconstruction) of our study have the following two data requirements. First, for examining effect of age diversity on spatial variability, we require surveys containing age-specific catch data on each stat square for each species. Second, the survey should continue for a sufficiently long interval for conducting CCM and S-map (time-series length $n > 30$ at least, and longer is better; Sugihara *et al.* 2012 *Science*). To this end, we explored all long-term surveys containing age-specific spatial catch data in “Exchange data” of DATRAS and found some other species that might be suitable for our study. Detailed information follows:

Survey	Species	N	Period	Quarter	Gear
NS-IBTS	Glyptocephalus cynoglossus	30	2004-2018	Q1 & Q3	GOV
NS-IBTS	Lophius piscatorius	31	2003-2018	Q1 & Q3	GOV
NS-IBTS	Merluccius merluccius	30	2004-2018	Q1 & Q3	GOV
NS-IBTS	Microstomus kitt	30	2004-2018	Q1 & Q3	GOV
NS-IBTS	Scophthalmus maximus	30	2004-2018	Q1 & Q3	GOV
BITS	Platichthys flesus	34	2002-2018	Q1 & Q4	TVL
BITS	Platichthys flesus	38	2000-2018	Q1 & Q4	TVS
BITS	Solea solea	30	2004-2018	Q1 & Q4	TVS

Then, we processed the data to delete empty stat squares throughout the survey period and removed time points containing less than 10 stat squares (the necessary criterion

for analyses, as explained in Methods in the main text). Unfortunately, none of the species listed in the above table satisfies our data criteria. Finally, we identified that only lemon sole (*Microstomus kitt*) had relatively long time-series data after data processing ($n = 25$, albeit n was still < 30 and much less than time-series length of IBTS data we used, with $n = 50$). Such a short time series will cause difficulty in state-space reconstruction and lead to a deficient attractor. Nevertheless, we still tried to apply CCM and S-map analyses on lemon sole. To accommodate the issue of short time series, we tried time lags up to four quarters for each variable. Thanks to Dr. Needle, who provided age-length key for lemon sole, which facilitated the analysis. Results are listed below.

Species	Dimensionality (E)	Library variable: spatial CV of CPUE				CV of SBT
		Age diversity	Abundance	AMO	SBT	
Microstomus kitt (lemon sole)	7	n.s.	n.s.	n.s.	0.070 (4)	n.s.

The CCM results indicated that only the sea bottom temperature (SBT) was detected as causal variable of spatial variability for lemon sole. However, the embedding dimension ($E = 7$) of spatial variability was much larger than the number of candidate causal variables. We were thus unable to reconstruct the attractor of spatial variability using any combination of causal variables in S-map analysis. The non-significant CCM result was likely due to the short time series that created difficulty in depicting the attractor (i.e. vector points were not dense enough to render the topology of attractor in a lagged-coordinate embedding space). Considering the potential issue of justifiability in the results of lemon sole and the time-scale discrepancy between the data of lemon sole and standard species, we decided not to include lemon sole into our study.

We have added more details in Methods explaining how we parsed and filtered surveys in DATRAS. Codes for parsing, filtering and manipulating data are also made publicly available.

Response 1: I'm afraid I don't accept this explanation, and I return to my concern that the authors are using data for stocks for which they are not suited. An assumption of constant bottom fraction does not hold for mackerel, which are very pelagic and whose position in the water column is certainly not fixed. For herring and sprat, they could have used the acoustic data which is also available (albeit in a beta form)

through the ICES website. For plaice, they should use the beam trawl survey (BTS) data which covers a large proportion of the plaice stock, uses gear designed for plaice, and is available from DATRAS. Saithe in the North Sea will always be difficult to model using North Sea surveys, as only a small fraction of the stock actually is found in the North Sea. I am concerned about the validity of the authors' very clever modelling for these species for this reason.

Response 1:

We agree that IBTS may not be adequate for some species and have explored other surveys as suggested. For herring and sprat, we have explored acoustic data available on the ICES website, but found that the acoustic survey was conducted annually from 2008. Thus, the time series was too short for our analysis ($n < 10$). For plaice, we have parsed all surveys in the beam trawl survey (BTS) and compiled the per-stat-square data separately for each survey, as no standardization of fishing gears across surveys exists. However, the time series was not long enough for each survey in BTS ($n = 26$ for Belgium and 23 for Netherlands; others were even shorter). Also, BTS and IBTS overlapped a lot in their spatial coverage (ICES division IVb and IVc). Considering that the time series of BTS are short and that IBTS covers a larger geographical range than each survey in BTS, we decided to use IBTS in our study for the validity and robustness of our analyses.

We agree that for some species (e.g. plaice, saithe and Atlantic mackerel), populations in the North Sea may not be representative for the whole stock. In particular, the reviewer suggested that Atlantic mackerel should not be included in the analysis, because only relatively young individuals were mainly caught in IBTS. This suggestion is based on the consideration that the IBTS data do not represent the whole population for the Atlantic mackerel (a consideration that is critical for stock assessment). However, our study does not require investigation on absolute changes in spatial structure of the whole stock, neither to estimate the actual population abundance. Rather, our study explored relative changes in spatial structure in response to changing age structure, abundance and environment. Please note that the data of spatial structure, age structure and abundance all come from the same IBTS dataset. As long as the data collected by the survey are not excessively scant, those sampled populations are still informative and should possess statistical meanings. Thus, even though populations in the North Sea are not representative for the whole stock, our results still illustrated dynamical coupling among these variables for that proportion of the population we examined. In addition, given that IBTS has been proceeding for a long period of time and has a large spatial coverage, growing literature has utilized

IBTS to character population spatial structure of pelagic species (including Atlantic mackerel) (Alheit *et al.* 2012 *Prog. Oceanogr.*; Peck *et al.* 2013 *Prog. Oceanogr.*; Montero-Serra *et al.* 2015 *Global Change Biol.*). Those studies support our arguments. Therefore, we disagree that Atlantic mackerel should be removed from the analyses. Nevertheless, we have explicitly pointed out potential issues of using IBTS for Atlantic mackerel and other pelagic species and clarified limitation of our study in Methodological considerations (**P13: L301-313**).

Response 2: The point here is that this is the case for just 2 out of 4 significant cases, which does not justify the statement made about the effect of age diversity.

Response 2:

We agree that the conclusion may be too assertive in the case for only two species showing significant negative age-diversity effect in S-map (actually it was three species if we include additional S-map analyses with alternative variable combinations; see results of plaice in Supplementary Table 3). However, the statement “2 out of 4 species” is not precise for explaining our results. In fact, the point here is that we detected age diversity negatively affect population spatial variability in ALL successfully reconstructed attractors in which age diversity is involved. That means, once we can successfully reconstruct the attractor using a combination of causal variables including age diversity, influential strengths of age diversity on population spatial variability are always negative, at least for all cases in our study. Unfortunately, we could only successfully reconstruct the attractor for three species (cod, plaice and mackerel) when including age diversity as a causal variable. However, non-significant S-map results for other species do NOT invalidate the (potentially) negative effect of age diversity, as CCM (Table 1) showed that age diversity affected population spatial variability for all species significantly. Instead, the non-significant S-map results only indicated that using the current combination of causal variables with age diversity could not reconstruct the attractor of population spatial variability. Therefore, there are other important factors affecting population spatial variability not captured in our study. We have clarified it in “Methodological considerations” and revised our conclusions to avoid over-explanation.

Response 3: I'm pleased to see this new section, I think it strengthens the paper a good deal.

Response 3:

Thank you.

Response 5: But if spatial distribution has changed over time, wouldn't that affect your conclusions about preferred habitat?

Response 5:

In theory, the spatial distribution of each age class might change with living conditions of habitats. Even so, based on our figures, each age class had its own preferred habitat over time (Fig. 3 and Supplementary Fig. 2). The proposed condition did not occur in our case.

Response 10: Again, I think this is overplaying it - the results were significant and positive for much less than "almost all" species examined.

Response 10:

We would like to clarify that we are not intended to suggest “all” species here and apologize for the wording if it causes confusion. Although our S-map results were only significant for a few species for reasons mentioned in Response 2, our CCM results indicated that variables examined in this study affected spatial variability for most species. Therefore, we are confident that variables examined (age diversity, abundance and environment) causally affect spatial variability for most species, though only for a few species we could determine the direction and strength of such effects. We have revised the manuscript (**P13-14: L315-345**) to make this more clear.

We have also revised our conclusions to be more conservative:

(P15: L360-367) Our findings highlighted potentially detrimental effects of fishing on population spatial structure, which may be associated with truncated age structure as well as diminished abundance. In addition, warming and fluctuating temperatures also had roles in undermining population age structure, abundance and spatial structure. Our study provided new knowledge to understand the causal network of population spatial structure. We emphasize the importance of population spatial structure to fishery management, particularly on intertwining effects of changes in population dynamics induced by fishing and environmental variability.

Response 11: See my comment above on the Exchange data format, which I think would be much more useful for these analyses (and cover many more relevant species).

Response 11:

Please see the Response to Summary.

Reviewer #3:

The authors have done a thorough job of addressing these comments. I have no other issues with the manuscript.

Response:

Thank you.